# TPP-LLM: Modeling Temporal Point Processes by Efficiently Fine-Tuning Large Language Models

## Abstract

Temporal point processes (TPPs) are widely used to model the timing and occurrence of events in domains such as social networks, transportation systems, and e-commerce. In this paper, we introduce TPP-LLM, a novel framework that integrates large language models (LLMs) with TPPs to capture both the semantic and temporal aspects of event sequences. Unlike traditional methods that rely on categorical event type representations, TPP-LLM directly utilizes the textual descriptions of event types, enabling the model to capture rich semantic information embedded in the text. While LLMs excel at understanding event semantics, they are less adept at capturing temporal patterns. To address this, TPP-LLM incorporates temporal embeddings and employs parameter-efficient fine-tuning (PEFT) methods to effectively learn temporal dynamics without extensive retraining. This approach improves both predictive accuracy and computational efficiency. Experimental results across diverse real-world datasets demonstrate that TPP-LLM outperforms state-of-the-art baselines in sequence modeling and event prediction, highlighting the benefits of combining LLMs with TPPs.

## 1 Introduction

Temporal point processes (TPPs) (Shchur et al., 2021) are powerful tools for modeling the occurrence of events over time, with widespread applications in domains such as social networks, urban dynamics, transportation, natural disasters, and e-commerce. The challenge of predicting both the type and timing of future events based on historical sequences has led to the development of increasingly sophisticated models. Traditional TPP models often rely on handcrafted features or specific assumptions about temporal dependencies, which limit their ability to capture complex event patterns in real-world datasets. Recent advances, such as neural TPPs, have leveraged the representational power of deep learning to overcome some of these limitations, but many still require extensive task-specific training from scratch.

With the rise of powerful large language models (LLMs), such as GPT-4 (Achiam et al., 2023) and Llama-3 (Dubey et al., 2024), new opportunities have emerged for using LLMs to understand and predict event sequences by capturing rich semantic and contextual information. Inspired by their success in text-based tasks (Zhao et al., 2023) and time series prediction (Zhou et al., 2023; Jin et al., 2023a; Zhang et al., 2024b), we propose TPP-LLM (Figure 1), a novel framework that integrates LLMs with TPPs to model both the temporal and semantic aspects of event sequences. By leveraging pretrained LLMs, TPP-LLM directly utilizes textual descriptions of event types, moving beyond traditional methods that rely on categorical representations. To ensure the model captures temporal dynamics, we incorporate temporal embeddings alongside these event descriptions. To efficiently adapt LLMs for TPP modeling, we employ low-rank adaptation (LoRA) (Hu et al., 2021), a parameter-efficient fine-tuning (PEFT) (Liu et al., 2022) method, allowing us to adjust a small subset of LLM parameters, reducing computational cost while maintaining high performance. Through extensive experiments on real-world datasets, we demonstrate that TPP-LLM consistently outperforms state-of-the-art baselines in sequence modeling and next event prediction.

The main contributions of this paper are as follows: (1) We introduce a novel approach that integrates LLMs with TPPs to improve event sequence modeling by leveraging textual event descriptions and

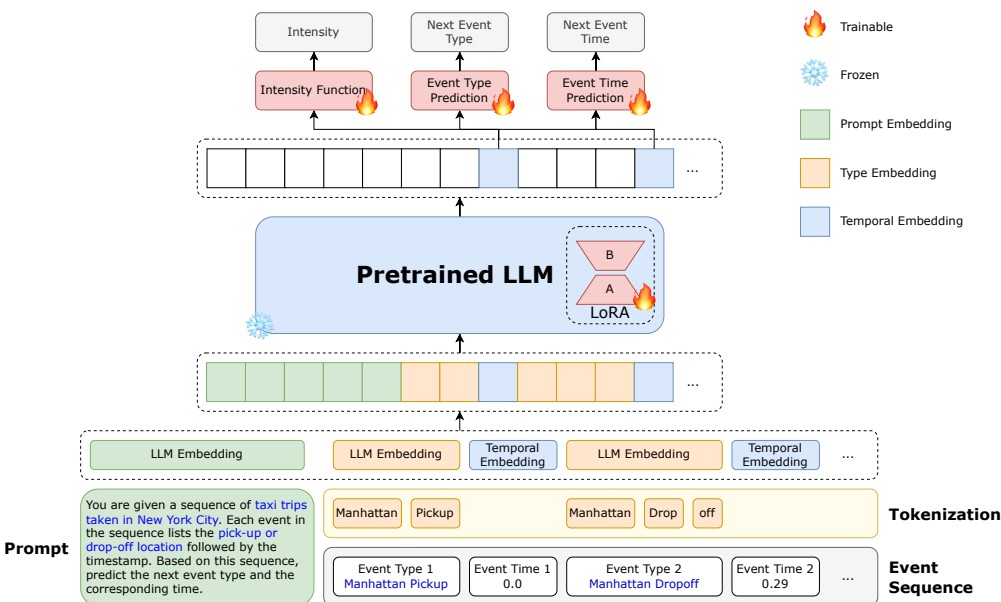

Figure 1: The TPP-LLM framework for event sequence prediction. Textual event descriptions are tokenized and processed through a pretrained LLM to capture semantic information, while temporal embeddings represent event timings. These are combined and passed through the LLM to generate history vectors. Low-rank adaptation (LoRA) optimizes the model for event sequences, with a trainable intensity function and head layers for predicting next events.

temporal embeddings. (2) We demonstrate the effectiveness of PEFT for modeling TPPs, allowing TPP-LLM to adapt pretrained LLMs without the need for full model retraining from scratch. (3) We conduct extensive experiments on multiple real-world datasets, showing that TPP-LLM achieves superior performance compared to existing neural TPP models. In the following sections, we discuss the related work, describe our methodology in detail, present the experimental results, and conclude with future directions for research.

## 2 RELATED WORK

**Neural Temporal Point Processes.** Recent advances in neural temporal point processes (TPPs) have introduced models that leverage deep learning techniques to capture complex temporal dependencies and event interactions. Many of these models use recurrent neural networks (RNNs) (Hochreiter, 1997) or self-attention mechanisms (Vaswani et al., 2017) to model event intensities based on event history. For example, RMTPP (Du et al., 2016) and NHP (Mei & Eisner, 2017) use RNNs to learn temporal influences, while more recent approaches like SAHP (Zhang et al., 2020) and THP (Zuo et al., 2020) utilize self-attention to capture long-term dependencies. Other models, such as those based on fully neural networks (Omi et al., 2019), normalizing flows (Shchur et al., 2019), neural ordinary differential equations (ODEs) (Chen et al., 2020), attention mechanisms (Yang et al., 2022), diffusion processes (Yuan et al., 2023), meta learning (Bae et al., 2023), and Mamba models (Gao et al., 2024), offer flexible and high-fidelity modeling of discrete events in continuous time. These methods have significantly improved the performance of TPPs by modeling complex interactions and dynamic event relationships.

**Large Language Models for Event Sequences.** Recent work has explored integrating large language models (LLMs) into event sequence prediction tasks (Jin et al., 2023b). Shi et al. (2024) propose LAMP, a framework that leverages LLMs for abductive reasoning to improve event sequence prediction. Xue et al. (2023) introduce PromptTPP, which incorporates continual learning into neural temporal point processes to enable adaptive and efficient learning of streaming event sequences. Song et al. (2024) present LaTee, a model utilizing an amortized expectation-maximization frame-

work with logic trees as latent variables and a learnable GFlowNet to generate logic tree samples for more effective event reasoning.

## 3 PRELIMINARIES

In this section, we introduce the necessary background on temporal point processes and their extensions using neural networks for modeling complex event sequences.

### 3.1 TEMPORAL POINT PROCESSES

Temporal point processes (TPPs) (Hawkes, 1971; Laub et al., 2015) are a class of stochastic processes used to model the occurrence of discrete events over continuous time. A marked TPP extends this framework by associating each event with both a time of occurrence and a type (mark), making it highly applicable in domains where understanding both event types and their timing is critical.

In a marked TPP, a sequence of events over an observation window $[0, T]$ is represented as: $\mathcal{S} = \{(t_1, k_1), (t_2, k_2), \ldots, (t_n, k_n)\}$, where $t_i$ represents the time of the $i$-th event and $k_i \in \mathcal{K}$ represents the corresponding event type from a discrete set $\mathcal{K} = \{1, 2, \ldots, K\}$. The goal is to model the probability of the next event's time and type, given the history of previous events.

The key function in a TPP is the conditional intensity function $\lambda(t, k|\mathcal{H}_t)$, which defines the instantaneous rate at which an event of type $k$ occurs at time $t$, conditioned on the history $\mathcal{H}_t$. Formally, it is defined as:

$$\lambda(t, k|\mathcal{H}_t) = \lim_{\Delta t \to 0} \frac{\mathbb{E}[N(t + \Delta t) - N(t)|\mathcal{H}_t]}{\Delta t}, \tag{1}$$

where $\mathcal{H}_t = \{(t_j, k_j) : t_j < t\}$ represents the history of previous events up to time $t$, and $N(t)$ is the counting process representing the number of events that have occurred up to time $t$. This intensity function provides the expected number of events occurring in a small time interval $[t, t + \Delta t)$, conditioned on the past. The joint probability density $p(t, k|\mathcal{H}_t)$ represents the likelihood of the next event occurring at time $t$ with type $k$, conditioned on the history $\mathcal{H}_t$. It is expressed as: $p(t, k|\mathcal{H}_t) = \lambda(t, k|\mathcal{H}_t) \exp\left(-\int_{t_i}^{t} \sum_{k' \in \mathcal{K}} \lambda(s, k'|\mathcal{H}_s) \, ds\right)$, where the integral accounts for no events occurring between the last event at $t_i$ and the current time $t$, capturing both event timing and type dependencies.

To evaluate the fit of a TPP model to observed data, the log-likelihood function is commonly used. The log-likelihood of observing a sequence of events $\mathcal{S}$ under a marked TPP is given by:

$$\mathcal{L}(\mathcal{S}) = \sum_{i=1}^{n} \log \lambda(t_i, k_i|\mathcal{H}_{t_i}) - \int_0^T \sum_{k \in \mathcal{K}} \lambda(t, k|\mathcal{H}_t) \, dt, \tag{2}$$

where the first term sums over the observed events, and the second term integrates over time and all possible event types $k$ to account for the likelihood of no events occurring between observations.

### 3.2 NEURAL TEMPORAL POINT PROCESSES

Recent advances in TPPs have introduced neural-based models that leverage the representational power of deep learning to capture complex event sequences. These models typically parameterize the conditional intensity function $\lambda(t, k|\mathcal{H}_t)$ using neural networks, enabling them to learn both temporal dependencies and event type distributions directly from data.

In neural TPPs, for each event $(t_i, k_i)$, an embedding $\boldsymbol{e}_i \in \mathbb{R}^D$ is computed through embedding layers based on the event time $t_i$ and the event type $k_i$. The hidden state $\boldsymbol{h}_t$, which represents the history up to time $t$, is then updated based on the current event's embedding and the previous hidden state $\boldsymbol{h}_{i-1}$. This update can be formulated as: $\boldsymbol{h}_i = f_{\text{update}}(\boldsymbol{h}_{i-1}, \boldsymbol{e}_i)$, where $f_{\text{update}}$ is a neural network, often implemented as a recurrent neural network (RNN) (Hochreiter, 1997) or a more advanced attention-based mechanism (Vaswani et al., 2017). With the updated hidden state $\boldsymbol{h}_i$, the next event time $t_{i+1}$ and event type $k_{i+1}$ are sampled from the probability distribution conditioned on $\boldsymbol{h}_i$: $t_{i+1}, k_{i+1} \sim P(t_{i+1}, k_{i+1}|\boldsymbol{h}_i)$. Different neural TPP models employ various architectures for the state update function $f$. Early approaches (Du et al., 2016; Mei & Eisner, 2017) use RNNs to capture temporal dependencies between events, while more recent models (Zhang et al., 2020; Zuo

et al., 2020; Yang et al., 2022) replace the recurrent structure with attention-based layers, allowing for better long-range interactions. These neural-based methods enhance the flexibility of TPPs, learning event dependencies from complex datasets in a data-driven manner.

## 4 METHODOLOGY

In this section, we introduce our proposed framework, TPP-LLM, which leverages large language models (LLMs) to model temporal point processes (TPPs). TPP-LLM, illustrated in Figure 1, integrates pretrained LLMs to capture the semantic richness of event types and employs temporal embeddings to handle the temporal dynamics of event sequences.

### 4.1 EVENT AND PROMPT EMBEDDINGS

TPP-LLM models the sequence of events $\mathcal{S} = \{(t_1, k_1), (t_2, k_2), \ldots, (t_n, k_n)\}$, where each event $e_i$ consists of a time $t_i$ and a corresponding event type $k_i$. Unlike conventional TPP models, which use discrete event types, TPP-LLM directly processes the textual descriptions of event types using a pretrained LLM. This enables the model to capture richer semantic information from the event text while learning temporal dependencies.

The event type $k_i$ is represented as a sequence of tokens. Let $x_i = \{x_{i,1}, x_{i,2}, \ldots, x_{i,L_i}\}$ be the sequence of tokens for event type $k_i$, where $L_i$ is the length of the tokenized event type. Each token $x_{i,j}$ is mapped to an embedding $\boldsymbol{x}_{i,j} \in \mathbb{R}^D$ through the pretrained LLM's embedding layer $\boldsymbol{E} \in \mathbb{R}^{V \times D}$, where $V$ is the vocabulary size and $D$ is the embedding dimension. In addition to the event type representation, TPP-LLM incorporates a temporal embedding to capture the time dynamics. Each event time $t_i$ is mapped to a temporal embedding $\boldsymbol{t}_i \in \mathbb{R}^D$ using an embedding layer: $\boldsymbol{t}_i = f_{\text{temporal}}(t_i)$, where $f_{\text{temporal}}$ can be a linear layer or a positional encoding. In this research, we utilize the temporal positional encoding (Zuo et al., 2020):

$$[\boldsymbol{t}_i]_j = \begin{cases} \cos\left(\frac{t_i}{10000^{(j-1)/D}}\right), & \text{when } j \text{ is odd,} \\ \sin\left(\frac{t_i}{10000^{j/D}}\right), & \text{when } j \text{ is even.} \end{cases} \tag{3}$$

Other temporal encoding methods (Zhang et al., 2020; Gao & Dai, 2024) can also be applied.

To model the joint dynamics of event types and their timing, we combine the event type representation $\boldsymbol{X}_i = [\boldsymbol{x}_{i,1}, \boldsymbol{x}_{i,2}, \ldots, \boldsymbol{x}_{i,L_i}] \in \mathbb{R}^{L_i \times D}$ with the temporal embedding $\boldsymbol{t}_i \in \mathbb{R}^D$. The concatenated representation for each event $(t_i, k_i)$ is given by: $\boldsymbol{E}_i = [\boldsymbol{x}_{i,1}, \boldsymbol{x}_{i,2}, \ldots, \boldsymbol{x}_{i,L_i}, \boldsymbol{t}_i]$ or $[\boldsymbol{t}_i, \boldsymbol{x}_{i,1}, \boldsymbol{x}_{i,2}, \ldots, \boldsymbol{x}_{i,L_i}] \in \mathbb{R}^{(L_i+1) \times D}$, depending on the event type and time order.

In addition to the event-specific embeddings, we also prepend a prompt as a sequence of tokens, which is similarly transformed into embeddings via the LLM's embedding layer: $\boldsymbol{P} = [\boldsymbol{p}_1, \boldsymbol{p}_2, \ldots, \boldsymbol{p}_{L_p}] \in \mathbb{R}^{L_p \times D}$. The prompt embeddings, along with the concatenated event type and temporal embeddings, form a unified sequence of embeddings: $\boldsymbol{X} = [\boldsymbol{P}, \boldsymbol{E}_1, \boldsymbol{E}_2, \ldots, \boldsymbol{E}_n] \in \mathbb{R}^{(L_p + \sum_i L_i + n) \times D}$, where $\boldsymbol{P}$ represents the prompt embeddings and $\boldsymbol{E}_i$ represents the event type and time embeddings of one event.

### 4.2 HISTORY VECTORS AND INTENSITY FUNCTION

The entire sequence $\boldsymbol{X}$ is then passed through the decoder-only transformer (LLM) to obtain contextualized hidden states for each token: $\boldsymbol{H} = \text{LLM}(\boldsymbol{X})$. After processing, we extract the hidden states corresponding to the last embedding vector of each event. For example, the hidden state of event $i$ is $\boldsymbol{h}_i = \boldsymbol{H}_{L_P + \sum_{j \leq i} L_j + i} \in \mathbb{R}^H$. The selected hidden state $\boldsymbol{h}_i$ represents the event history up to time $t_i$: $\mathcal{H}'_{t_i} = \{(t_j, k_j) : t_j \leq t_i\}$. These history vectors are then used for modeling TPPs.

In our model, the intensity function is parameterized using the history vector $\boldsymbol{h}_i$, which encodes the event history from the initial time to time $t_i$. To compute the intensity function between $t_i$ and $t_{i+1}$, we apply the linear transformation to the hidden state $\boldsymbol{h}_i$. For the event type $k$, the intensity function (Du et al., 2016; Zuo et al., 2020; Gao & Dai, 2024) is modeled as:

$$\lambda_k(t|\mathcal{H}_t) = \lambda(t, k|\mathcal{H}_t) = f_k(\alpha_k(t - t_i) + \boldsymbol{w}_k^\mathsf{T} \boldsymbol{h}_i + b_k), \tag{4}$$

where $f_k = \log(1 + \exp(x))$ is the softplus function, $\alpha_k \in \mathbb{R}$, $\boldsymbol{w}_k \in \mathbb{R}^H$, and $b_k \in \mathbb{R}$ are the learnable parameters. The softplus activation ensures the intensity function is non-negative.

### 4.3 EVENT PREDICTION

For each event $(t_i, k_i)$, the history vector $\boldsymbol{h}_i$ from the LLM output encodes the event history $\mathcal{H}_{t_i}$, which includes both the event type and temporal dynamics up to time $t_i$. Following previous research (Zuo et al., 2020; Gao & Dai, 2024), we utilize this hidden representation to predict both the next event type $k_{i+1}$ and time $t_{i+1}$ through separate layers. To predict the event type, we apply a linear layer followed by a softmax activation to the hidden state $\boldsymbol{h}_i$, mapping it to a probability distribution over the possible event types: $\hat{\boldsymbol{p}}_{i+1} = \hat{\boldsymbol{p}}(k_{i+1}|\mathcal{H}'_{t_i}) = \text{softmax}(\boldsymbol{W}_{\text{type}}\boldsymbol{h}_i + \boldsymbol{b}_{\text{type}})$, where $\boldsymbol{W}_{\text{type}} \in \mathbb{R}^{K \times H}$ and $\boldsymbol{b}_{\text{type}} \in \mathbb{R}^K$ are the weights and bias of the linear layer, $K$ is the number of event types, and $H$ is the hidden state dimension. The predicted event type $\hat{k}_{i+1}$ is predicted as the type with the maximum probability: $\hat{k}_{i+1} = \arg\max_k \hat{\boldsymbol{p}}_{i+1}$. Similarly, to predict the next event time, we apply another linear layer to the hidden state $\boldsymbol{h}_i$, producing a scalar value that represents the next time: $\hat{t}_{i+1} = \boldsymbol{w}_{\text{time}}^{\mathsf{T}}\boldsymbol{h}_i + b_{\text{time}}$, where $\boldsymbol{w}_{\text{time}} \in \mathbb{R}^H$ and $b_{\text{time}} \in \mathbb{R}$ are the weights and bias for this layer.

### 4.4 FINE-TUNING

To efficiently adapt the pretrained LLM to the TPP task, we employ low-rank adaptation (LoRA) (Hu et al., 2021), a parameter-efficient fine-tuning (PEFT) (Liu et al., 2022) method. Instead of fine-tuning all the parameters of the LLM, low-rank matrices are introduced to LLM weights. Specifically, we modify the weight matrix of one target module: $W' = W + BA$, where $W$ is the original weight, and $A$, $B$ are learnable low-rank matrices. By fine-tuning only these low-rank matrices, we significantly reduce the number of trainable parameters, making the adaptation more efficient without compromising performance. In addition to LoRA, other PEFT methods (Liu et al., 2022; Zhang et al., 2023) can also be applied to further optimize the fine-tuning process.

To fine-tune the LLM alongside the additional head layers, we define a combined loss function that includes the log-likelihood of observed events, event type prediction loss, and event time prediction loss. The likelihood function Equation 2 based on the conditional intensity function is adapted to:

$$\mathcal{L}(\mathcal{S}) = \sum_{i=1}^{n} \log \lambda(t_i, k_i|\mathcal{H}_{t_i}) - \int_{t_1}^{t_n} \sum_{k \in \mathcal{K}} \lambda(t, k|\mathcal{H}_t)\, \mathrm{d}t, \tag{5}$$

where the non-event integral can be computed by Monte Carlo or numerical integration methods (Zuo et al., 2020). The event type loss is defined as the cross-entropy between true and predicted event types: $\mathcal{L}_{\text{type}}(\mathcal{S}) = \sum_{i=2}^{n} -\boldsymbol{k}_i^{\mathsf{T}} \log(\hat{\boldsymbol{p}}_i) = \sum_{i=2}^{n} -\log([\hat{\boldsymbol{p}}_i]_{k_i})$, where $\boldsymbol{k}_i$ is the one-hot encoding for the ground-truth $k_i$. The event time loss is defined as the mean squared error between true and predicted event times: $\mathcal{L}_{\text{time}}(\mathcal{S}) = \sum_{i=2}^{n} (t_i - \hat{t}_i)^2$. The training objective is defined as the sum of the negative log-likelihood, along with the event type and time losses, over all sequences $\mathcal{S}_i$:

$$\sum_{i=1}^{N} \ell(\mathcal{S}_i) = \sum_{i=1}^{N} \left( -\mathcal{L}(\mathcal{S}_i) + \beta_{\text{type}}\mathcal{L}_{\text{type}}(\mathcal{S}_i) + \beta_{\text{time}}\mathcal{L}_{\text{time}}(\mathcal{S}_i) \right), \tag{6}$$

where $\beta_{\text{type}}$ and $\beta_{\text{time}}$ are coefficients for the event type and time losses.

## 5 EXPERIMENTS

In this section, we present the experimental evaluation of our proposed TPP-LLM model. We detail the datasets, prompts used, baseline models, experimental settings, results, and ablation analysis.

### 5.1 DATASETS

We conduct experiments on five real-world datasets: Stack Overflow, Chicago Crime, NYC Taxi Trip, U.S. Earthquake, and Amazon Review. Their statistics are shown in Table 1. The datasets span various applications and are widely used in prior TPP research, making them well-suited for evaluating the performance of our model. However, since the currently available versions lack the corresponding event type texts required by TPP-LLM, we preprocess data to include these critical textual descriptions. These diverse datasets allow us to assess the model's generalization capabilities across different domains, handling sequences with varying lengths, event types, and temporal resolutions. More detailed information is available in Appendix A.

Table 1: Dataset statistics overview for event sequences.

| Dataset | # of Types | # of Events | # of Seq. | Avg. Seq. Length | Time Unit |
|---|---|---|---|---|---|
| Stack Overflow | 25 | 187,836 | 3,336 | 56.31 | Month |
| Chicago Crime | 20 | 202,333 | 4,033 | 50.17 | Month |
| NYC Taxi Trip | 8 | 362,374 | 2,957 | 122.55 | Hour |
| U.S. Earthquake | 3 | 29,521 | 3,009 | 9.81 | Day |
| Amazon Review | 18 | 127,054 | 2,245 | 56.59 | Week |

## 5.2 PROMPT DESIGN

We design the prompt to provide a structured guide for the model, helping it understand the task and the sequence of events effectively. The prompt includes essential details such as the sequence context and specifics about event types, allowing the model to focus on the key components it needs to process for accurate predictions. The general structure of the prompt is as follows: "{Sequence Description} {Event Description} {Task Description}" with the task description tailored to the prediction task. When event type precedes time in the embedding sequences, the task is framed as: "Based on this sequence, predict the next event type and the corresponding time." Alternatively, when event time comes first, the task becomes: "Based on this sequence, predict the next event time and the corresponding type." The specific sequence and event descriptions for datasets used in our experiments are listed in Appendix B.

## 5.3 BASELINES AND EVALUATION METRICS

We compare our model, TPP-LLM, with several state-of-the-art (SOTA) baselines to evaluate its performance across different tasks. The baselines include the Neural Hawkes Process (NHP) (Mei & Eisner, 2017), Self-Attentive Hawkes Process (SAHP) (Zhang et al., 2020), Transformer Hawkes Process (THP) (Zuo et al., 2020), Attentive Neural Hawkes Process (AttNHP) (Yang et al., 2022), Neural ODE-based Temporal Point Process (ODETPP) (Chen et al., 2020), and Meta Temporal Point Process (MetaTPP) (Bae et al., 2023). These models represent leading approaches in neural TPP modeling. Detailed descriptions of the baselines are provided in Appendix C.

To assess model performance, we use the following evaluation metrics: The **log-likelihood** measures how well the model fits the observed event sequence $\mathcal{S}$, which is computed as Equation 5 with the intensity function. **Accuracy** is used to evaluate the event type prediction, measuring the proportion of correctly predicted event types: $\text{Accuracy} = \frac{1}{n} \sum_{i=1}^{n} \mathbb{1}(k_i = \hat{k}_i)$, where $k_i$ is the true event type, $\hat{k}_i$ is the predicted event type, and $\mathbb{1}$ is the indicator function. **Root mean squared error (RMSE)** is used to measure the error in predicting the event times. It is calculated as: $\text{RMSE} = \sqrt{\frac{1}{n} \sum_{i=1}^{n} (t_i - \hat{t}_i)^2}$, where $t_i$ is the true event time and $\hat{t}_i$ is the predicted event time.

## 5.4 EXPERIMENTAL SETUP

We conduct experiments using two foundation models for TPP-LLM: TinyLlama-1.1B-Chat-v1.0 (Zhang et al., 2024a) and Gemma-2-2B-IT (Team et al., 2024), both of which are quantized to 4-bit precision (Dettmers et al., 2024) for efficient GPU memory usage. To capture temporal dynamics, we use temporal positional encoding (Zuo et al., 2020), and event type embeddings are processed first, followed by the temporal embedding for each event. The non-event integral term in the log-likelihood is handled using Monte Carlo integration (Zuo et al., 2020) with 20 samples per time interval, applied consistently across all models. For fine-tuning, we employ LoRA (Hu et al., 2021) by adapting weight matrices in attention modules, with dropout applied but without bias. The Adam optimizer (Kingma, 2014) is used for optimizing both the LoRA layers and prediction layers. Baselines implemented in the EasyTPP framework (Xue et al., 2024) are utilized, with hyperparameters adapted from it for a fair comparison. Experiments results are averaged over five runs with early stopping, and additional hyperparameters and setup are provided in Appendix D and E. We utilize a single NVIDIA A10 and A100 GPU for baselines and a single H100 GPU for TPP-LLM.

## 5.5 EXPERIMENTAL RESULTS

We evaluate TPP-LLM against baselines across five real-world datasets. Two TPP-LLM models are included: TPP-Llama (TinyLlama-1.1B-Chat-v1.0) and TPP-Gemma (Gemma-2-2B-IT).

**Log-Likelihood Performance.** In terms of log-likelihood (Table 2), TPP-LLM models (TPP-Llama and TPP-Gemma) demonstrate competitive performance across most datasets. TPP-Llama achieves the best performance on Stack Overflow, while AttNHP outperforms all models on Chicago Crime, NYC Taxi Trip, and Amazon Review. However, TPP-LLM models still perform strongly, with ranking second on most datasets, except for U.S. Earthquake, where SAHP achieves the top score. These results highlight TPP-LLM's ability to model complex event sequences effectively, particularly benefiting from the LLM's ability to capture event semantics. Despite being outperformed on some datasets, TPP-LLM models remain highly competitive overall.

Table 2: Performance comparison of log-likelihood across different datasets.

| Model | StackOverflow | Crime | Taxi | Earthquake | Amazon |
|---|---|---|---|---|---|
| NHP | -2.005 | -2.604 | 0.366 | -0.450 | -1.196 |
| SAHP | -6.320 | -6.069 | -0.228 | **0.193** | -4.201 |
| THP | -1.877 | -2.493 | 0.217 | -0.513 | -1.083 |
| AttNHP | -1.798 | **-2.432** | **0.446** | -0.481 | **-0.959** |
| ODETPP | -2.402 | -4.152 | -0.450 | -0.511 | -1.808 |
| MetaTPP | -3.006 | -2.996 | -1.268 | -0.649 | -5.161 |
| TPP-Llama | **-1.777** | -2.451 | 0.271 | -0.475 | -1.011 |
| TPP-Gemma | -1.785 | -2.480 | 0.332 | -0.479 | -1.075 |

**Event Type Prediction Accuracy.** For next event type prediction accuracy (Table 3 and Figure 2), TPP-LLM outperforms or matches the performance of baselines across all datasets. TPP-Llama achieves the highest accuracy on Stack Overflow and Amazon Review, while TPP-Gemma excels on NYC Taxi Trip and U.S. Earthquake. Both variants demonstrate substantial improvements over other baselines, particularly when dealing with datasets like Stack Overflow and Amazon Review, where rich event-type semantics can be leveraged by LLMs to improve prediction accuracy. This highlights TPP-LLM's capacity to integrate event text information into the prediction process, providing a clear advantage over traditional TPP models.

Table 3: Performance comparison of next event type prediction accuracy and event time prediction RMSE across different datasets.

| Model | StackOverflow | Crime | Taxi | Earthquake | Amazon |
|---|---|---|---|---|---|
| NHP | 42.18%/0.629 | 25.20%/0.736 | 90.78%/0.960 | 62.58%/0.389 | 65.97%/0.721 |
| SAHP | 38.63%/0.588 | 21.39%/0.691 | 88.02%/0.881 | 60.11%/**0.271** | 65.93%/0.662 |
| THP | 43.81%/0.629 | 26.70%/0.745 | 90.85%/0.924 | 62.39%/0.377 | 68.56%/0.733 |
| AttNHP | 39.12%/0.581 | **26.97%**/0.679 | 83.76%/0.904 | 61.87%/0.386 | 68.90%/0.658 |
| ODETPP | 40.33%/0.693 | 18.56%/0.848 | 86.63%/0.896 | 61.49%/0.490 | 65.39%/0.941 |
| MetaTPP | 41.06%/3.853 | 8.19%/2.037 | 45.26%/6.341 | 62.48%/0.515 | 65.30%/18.567 |
| TPP-Llama | **44.20%**/0.477 | 26.86%/**0.562** | 91.37%/0.884 | 62.70%/0.288 | **69.22%**/0.580 |
| TPP-Gemma | 43.94%/**0.474** | 24.54%/0.565 | **91.46%**/**0.840** | **63.12%**/0.286 | 67.71%/**0.578** |

**Event Time Prediction RMSE.** When evaluating next event time prediction (Table 3 and Figure 3), TPP-LLM once again delivers strong results. TPP-Gemma achieves the lowest RMSE on Stack Overflow, NYC Taxi Trip, and Amazon Review , while TPP-Llama performs best on Chicago Crime. Both variants significantly outperform baselines, particularly in datasets like Stack Overflow, Chicago Crime, and Amazon Review, where temporal patterns are less regular. This suggests that the LLM-based temporal embeddings in TPP-LLM are effective at capturing temporal dynamics, leading to more accurate event time predictions.

Overall, TPP-LLM demonstrates strong and consistent performance across all datasets. The inclusion of LLMs for event text processing and understanding allows the model to utilize richer contextual information, leading to better event type prediction accuracy. Additionally, the integration of temporal embeddings helps capture complex temporal dependencies, reflected in the model's strong

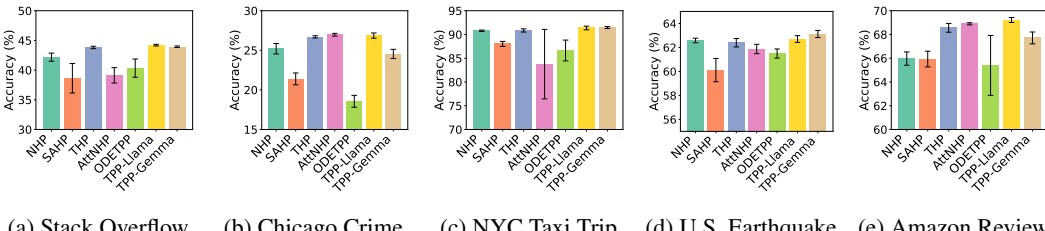

Figure 2: Performance comparison of next event type prediction accuracy across five different datasets. Each subplot shows the accuracy of models with error bars.

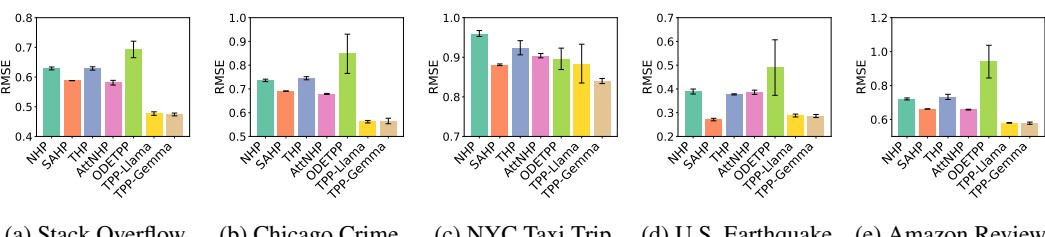

Figure 3: Performance comparison of next event time prediction RMSE across five different datasets. Each subplot shows the RMSE of models with error bars.

RMSE performance for event time predictions. The results confirm that TPP-LLM is an effective and adaptable model for various TPP tasks, achieving leading performance in real-world scenarios.

## 5.6 FEW-SHOT LEARNING

In the few-shot experiments using only 2% of the training data, TPP-LLM models (TPP-Llama and TPP-Gemma) perform strongly across datasets. For log-likelihood (Table 4), TPP-Llama excels on Stack Overflow and Amazon Review, while TPP-Gemma leads on NYC Taxi Trip. AttNHP performs best on Chicago Crime and U.S. Earthquake, with TPP-Llama remaining competitive. In terms of next event type accuracy (Table 5), TPP-Gemma dominates on Stack Overflow, NYC Taxi Trip, and Amazon Review, while TPP-Llama tops U.S. Earthquake. Both TPP-LLM models significantly outperform baselines like NHP and SAHP. For next event time RMSE (Table 5), TPP-Gemma leads on Stack Overflow and Chicago Crime, with SAHP and NHP showing competitive results on other datasets. These findings highlight TPP-LLM's strong adaptability in few-shot scenarios, effectively leveraging pretrained knowledge.

Table 4: Performance comparison of log-likelihood across different datasets on the 2% training set.

| Model | StackOverflow | Crime | Taxi | Earthquake | Amazon |
|---|---|---|---|---|---|
| NHP | -6.515 | -6.202 | -1.236 | -0.697 | -4.106 |
| SAHP | -6.839 | -6.301 | -1.173 | -0.615 | -4.567 |
| THP | -2.683 | -2.926 | -0.878 | -0.625 | -1.768 |
| AttNHP | -2.070 | **-2.594** | -0.670 | **-0.521** | -1.474 |
| ODETPP | -3.517 | -4.370 | -1.702 | -0.677 | -4.539 |
| TPP-Llama | **-2.026** | -2.668 | -0.098 | -0.539 | **-1.469** |
| TPP-Gemma | -2.068 | -2.716 | **0.051** | -0.600 | -1.630 |

## 5.7 ABLATION STUDIES

To understand the contribution of different components in TPP-LLM, we conduct a series of ablation studies. By systematically removing or altering key parts of the model, we analyze how each element affects overall performance and identify which configurations lead to the best results.

Table 5: Performance comparison of next event type prediction accuracy and event time prediction RMSE across different datasets on the 2% training set.

| Model | StackOverflow | Crime | Taxi | Earthquake | Amazon |
|---|---|---|---|---|---|
| NHP | 35.23%/0.587 | 8.27%/0.692 | 87.70%/0.898 | 60.36%/0.482 | 65.08%/**0.659** |
| SAHP | 37.12%/0.589 | 6.72%/0.692 | 45.53%/**0.895** | 59.84%/**0.374** | 65.33%/0.666 |
| THP | 35.51%/0.783 | 19.87%/0.875 | 46.15%/0.909 | 54.09%/0.388 | 65.33%/0.712 |
| AttNHP | **41.26%**/0.635 | **25.23%**/0.726 | 47.95%/0.911 | 61.70%/**0.374** | 65.33%/0.718 |
| ODETPP | 32.37%/0.863 | 18.82%/0.793 | 48.25%/1.426 | 60.70%/0.828 | 58.80%/1.650 |
| TPP-Llama | 41.07%/0.567 | 22.92%/0.647 | 90.13%/1.118 | **61.93%**/0.613 | 65.33%/0.833 |
| TPP-Gemma | **41.26%/0.514** | 20.84%/**0.604** | **90.58%**/1.103 | 61.84%/0.538 | **65.42%**/0.683 |

### 5.7.1 FOUNDATION MODELS

The performance comparison in Table 6 shows the impact of different LLMs on TPP-LLM's performance. TinyLlama-1.1B-Chat-v1.0 and TinyLlama-1.1B-Intermediate show similar log-likelihood and accuracy scores, with Chat slightly outperforming on next event type prediction for Stack Overflow and U.S. Earthquake. Gemma-2-2B-IT achieves the best RMSE for event time prediction on NYC Taxi Trip and U.S. Earthquake, highlighting its strength in modeling temporal dynamics. The Llama-3.2 models (Dubey et al., 2024) excel in log-likelihood for Stack Overflow and U.S. Earthquake, with Llama-3.2-1B-Instruct achieving the highest accuracy for NYC Taxi Trip and U.S. Earthquake, showcasing their strong performance across diverse metrics and datasets. Overall, the consistent performance across models underscores the robustness of TPP-LLM.

Table 6: Performance comparison of log-likelihood, next event type prediction accuracy, and next event time prediction RMSE across different datasets with various foundation models.

| Foundation Model | StackOverflow | Taxi | Earthquake |
|---|---|---|---|
| TinyLlama-1.1B-Intermediate | -1.777/44.17%/0.477 | 0.313/91.56%/0.847 | -0.476/62.65%/0.299 |
| TinyLlama-1.1B-Chat-v1.0 | -1.777/**44.20%**/0.477 | 0.271/91.37%/0.884 | -0.475/62.70%/0.288 |
| Gemma-2-2B | -1.787/44.09%/0.475 | **0.334**/91.52%/0.841 | -0.481/62.45%/0.299 |
| Gemma-2-2B-IT | -1.785/43.94%/**0.474** | 0.332/91.46%/**0.840** | -0.479/63.12%/**0.286** |
| Llama-3.2-1B | **-1.770**/44.12%/0.486 | 0.318/91.75%/0.875 | -0.472/62.86%/0.301 |
| Llama-3.2-1B-Instruct | -1.775/44.15%/0.483 | 0.310/**91.78%**/0.886 | -0.481/**63.14%**/0.303 |
| Llama-3.2-3B | -1.777/44.16%/0.491 | 0.285/91.69%/0.921 | **-0.466**/62.84%/0.312 |
| Llama-3.2-3B-Instruct | -1.771/44.19%/0.494 | 0.300/91.57%/0.883 | -0.469/63.02%/0.311 |

### 5.7.2 TEMPORAL EMBEDDINGS

As shown in Table 7, the type and order of temporal embeddings influence model performance. Temporal positional encoding generally outperforms both time-shifted positional encoding and linear embeddings in most cases. Specifically, when event time embeddings are processed first, temporal positional encoding yields the best next event type prediction accuracy and competitive RMSE values on Stack Overflow and U.S. Earthquake. Linear embeddings also show strong results, with the best log-likelihood on U.S. Earthquake when event time is placed first. Time-shifted positional encoding exhibits lower performance across all metrics. These findings suggest that processing the event time before the event type improves event type prediction, while adjusting the embedding strategy can optimize model performance for different metrics.

### 5.7.3 FINE-TUNING METHODS

Table 8 and Figure 4 illustrate the impact of different fine-tuning methods on performance. Without fine-tuning (only training the head layers), the model suffers significant drops in log-likelihood and accuracy, highlighting the need for adapting the pretrained LLM. Fine-tuning with LoRA consistently enhances performance, with higher ranks benefiting more complex tasks and lower ranks offering competitive results with less computational cost. Additionally, alternative methods like LoHa (Hyeon-Woo et al., 2022), LoKr (Yeh et al., 2023), and IA3 (Liu et al., 2022) demonstrate unique strengths: LoKr achieves excellent efficiency with the lowest trainable parameters, LoHa

Table 7: Performance comparison of log-likelihood, next event type prediction accuracy, and next event time prediction RMSE across different datasets with various temporal embeddings.

| Temporal Embedding | First Embedding | StackOverflow | Earthquake |
|---|---|---|---|
| Temporal Positional Encoding | Event Type | -1.777/44.20%/**0.477** | -0.475/62.70%/**0.288** |
| Temporal Positional Encoding | Event Time | **-1.763/44.47%**/0.477 | -0.475/**63.15%**/0.307 |
| Time-Shifted Positional Encoding | Event Type | -1.782/44.13%/0.550 | -0.475/62.97%/0.360 |
| Time-Shifted Positional Encoding | Event Time | -1.812/43.99%/0.556 | -0.474/63.11%/0.385 |
| Linear Layer | Event Type | -1.768/44.15%/0.478 | -0.474/62.65%/0.297 |
| Linear Layer | Event Time | -1.768/44.20%/**0.477** | **-0.467**/63.01%/0.343 |

shows strong log-likelihood on Stack Overflow, and IA3 performs well on RMSE for event type prediction. These results highlight trade-offs between computational efficiency and predictive performance among fine-tuning methods.

Table 8: Performance comparison of log-likelihood, next event type prediction accuracy, and next event time prediction RMSE across different datasets with various fine-tuning settings. (Trainable %: Percentages of trainable parameters in the foundation model, excluding prediction head layers.)

| Fine-Tuning | Quantization | Trainable | StackOverflow | Earthquake |
|---|---|---|---|---|
| None | 4-bit | - | -1.891/42.43%/0.484 | -0.497/62.95%/0.306 |
| LoRA (rank 4) | 4-bit | 0.109% | -1.774/44.18%/0.474 | -0.486/62.78%/0.291 |
| LoRA (rank 8) | 4-bit | 0.217% | -1.767/44.20%/0.484 | -0.480/63.19%/0.296 |
| LoRA (rank 16) | 4-bit | 0.434% | -1.777/44.20%/0.477 | -0.475/62.70%/**0.288** |
| LoRA (rank 32) | 4-bit | 0.864% | -1.771/**44.39%**/0.482 | -0.475/63.24%/0.292 |
| LoRA (rank 16) | - | 0.434% | -1.774/44.15%/0.480 | -0.475/62.84%/0.304 |
| LoHa (rank 8) | - | 0.434% | **-1.762**/44.23%/0.484 | -0.483/62.91%/0.301 |
| LoKr (rank 64) | - | 0.028% | **-1.762**/44.17%/0.499 | **-0.470**/63.13%/**0.288** |
| IA3 | - | 0.031% | -1.770/43.99%/**0.473** | -0.487/63.24%/0.293 |
| IA3 | 4-bit | 0.031% | -1.769/44.00%/0.477 | -0.484/**63.35%**/0.296 |

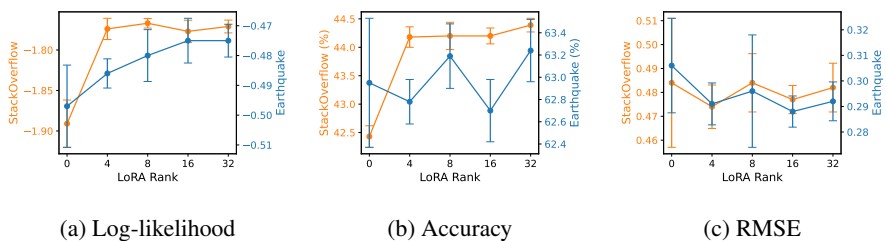

(a) Log-likelihood      (b) Accuracy      (c) RMSE

Figure 4: Performance comparison of log-likelihood, accuracy, and RMSE for different LoRA ranks with corresponding error bars across the Stack Overflow and U.S. Earthquake datasets.

## 6 CONCLUSION

In this paper, we introduced TPP-LLM, a novel framework for modeling temporal point processes (TPP) by leveraging the pretrained knowledge of large language models (LLMs). By integrating LLMs with temporal embeddings, our approach effectively captures both the event semantics and the temporal dynamics of complex event sequences. Through extensive experiments on real-world datasets, we demonstrated that TPP-LLM outperforms state-of-the-art baselines in terms of sequence modeling and next event prediction. Additionally, our ablation studies revealed the contributions of foundation models, temporal embeddings, prompt design, and fine-tuning strategies to overall performance. The robustness of TPP-LLM across diverse datasets and tasks highlights its potential for broader applications in TPP modeling. Future work could explore alternative fine-tuning techniques and embedding strategies, as well as extend this approach to multi-task settings.

## REPRODUCIBILITY STATEMENT

We have made several efforts to ensure the reproducibility of our findings by providing detailed documentation of the experimental setup and methodologies. Detailed descriptions of the datasets used in our experiments, including the preprocessing steps and sequence structure, can be found in Section 5.1 and Appendix A. The code used for implementing our TPP-LLM model, including the fine-tuning mechanisms, is made available as supplementary material. Model architecture details, training procedures, and hyperparameter settings, are provided in Section 5.4, Appendix D, and Appendix E, enabling the replication of experiments. Additionally, the theoretical foundations of our model are fully explained in Section 3 and 4. This comprehensive approach is intended to facilitate the reproduction of our results and to support further research building on our work.

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

# A    DATASET DETAILS

In this appendix, we provide additional information on the datasets used in our experiments, including detailed preprocessing steps and a breakdown of event types for each dataset.

Table 9: Numbers of sequences in train, validation, and test splits of datasets.

| Dataset | # of Seq. | # of Train Seq. | # of Val Seq. | # of Test Seq. |
|---|---|---|---|---|
| Stack Overflow | 3,336 | 2,668 | 334 | 334 |
| Chicago Crime | 4,033 | 3,226 | 403 | 404 |
| NYC Taxi Trip | 2,957 | 2,365 | 296 | 296 |
| U.S. Earthquake | 3,009 | 2,407 | 301 | 301 |
| Amazon Review | 2,245 | 1,796 | 224 | 225 |

## A.1    DATASET SUMMARIES

**Stack Overflow.** We use the badge subset from the Stack Overflow dataset, focusing on non-tag affiliated badges that can be awarded multiple times between January 1, 2022, and December 31, 2023. The dataset includes users with 40-100 badges and badges awarded at least 200 times, resulting in 3,336 sequences with 187,836 events and 25 event types.

**Chicago Crime.** The Chicago crime dataset covers incidents from January 1, 2022, to December 31, 2023. We focus on the top 20 primary crime types and blocks with 30-120 crime counts. This yields 4,033 sequences with 202,333 events and 20 event types.

**NYC Taxi Trip.** The NYC taxi dataset spans May 1-7, 2013, excluding trips from or to Staten Island. We keep sequences with 100-160 events, ensuring consecutive events occur within 12 hours. The final dataset contains 2,957 sequences with 362,374 events and 8 location types.

**U.S. Earthquake.** The United States Earthquake dataset includes earthquakes from January 1, 2020, to December 31, 2023, with events classified as "Large", "Medium", or "Small". We keep sequences with 5-30 events and maximum 24-hour time intervals. The dataset comprises 3,009 sequences with 29,521 events across 3 magnitude types.

**Amazon Review.** The Amazon review dataset includes reviews from January 1, 2018, to June 30, 2018. After combining same-day reviews in the same category, we focus on users with 40-200 category reviews across 17 categories (plus an "Other" category). This results in 2,245 sequences with 127,054 events across 18 category types.

## A.2    DATASET PREPOSSESSING

**Stack Overflow.** Stack Overflow is a popular question-answering website and online platform where developers and programmers ask and answer technical questions, share knowledge, and solve coding problems collaboratively. We select the badge subset of the Stack Overflow dataset[1], which includes user IDs, badge names, and others. For data preprocessing methods, we refer to the paper by Du et al. (2016). There are 94 types of non-tag affiliated badges as of March 31, 2024. These badges are designed to recognize a user's contributions and achievements within the community without being tied to specific tags or categories. We employ badges database schema to parse the data. We select data spanning from January 1, 2022, to December 31, 2023 and keep only the first record if there are duplicates for the same user at the same time due to technical issues. There are 39 badges that can be awarded multiple times by one user. We then select users who have earned between 40 badges and 100 badges and those badges that have been awarded at least 200 times to the selected users. We group the sequences by user. Finally, there are 3,336 sequences with 187,836 events and 25 event types.

**Chicago Crime.** Chicago crime dataset[2] includes reported crime incidents, excluding murders, that occurred in the City of Chicago. We remove records with missing values in the date, block, or

---

[1] https://archive.org/details/stackexchange

[2] https://data.cityofchicago.org/Public-Safety/Crimes-2001-to-Present/ijzp-q8t2

primary crime type fields. Then, we keep only the first record for duplicates with the same block, date, and primary crime type between January 1, 2022 and December 31, 2023. Next, we select the top 20 most frequently occurring primary crime types and choose blocks with crime counts between 30 and 120. We group the sequences by block. Finally, we obtain 4,033 sequences with 202,333 events and 20 event types.

**NYC Taxi Trip.** NYC taxi trip dataset[3] contains detailed records of taxi trips in New York City, including information such as pick-up and drop-off locations, times, and other relevant details. We first drop any records with missing values and remove duplicate entries, retaining only the first occurrence of each. Additionally, we exclude trips with zero longitude or latitude coordinates. We select data with pickup times spanning from May 1, 2013, to May 7, 2013, and exclude any trips originating from or ending in Staten Island. We divide the sequence based on hack license, ensuring that any two consecutive events are within 12 hours. We select sequences with event counts between 100 and 160. Finally, we obtain 2,957 sequences with 362,374 events and 8 location types.

**U.S. Earthquake.** The United States earthquake dataset[4] includes information on the time, latitude, longitude, and magnitude of earthquakes spanning from January 1, 2020 to December 31, 2023. We remove records with missing values in the time, coordinate (latitude and longitude), or magnitude fields, and then keep only the first record for duplicates with the same time, coordinate, and magnitude. We divide the sequence based on coordinates with nearest integers, ensuring that any two consecutive events are within 24 hours. If not, a new sequence is started. Then, we only keep the data with Richter magnitude scale (Local magnitude scale, ML) and select sequences with event counts between 5 and 30. We classify the magnitude into three categories, inspired by Zuo et al. (2020). When the magnitude is between 1 (inclusive) and 2 (exclusive), the event is classified as "Medium"; if the magnitude is greater than or equal to 2, it is classified as "Large." Magnitudes smaller than 1 are classified as "Small." In total, we identified 3,009 sequences consisting of 29,521 events across three magnitude types.

**Amazon Review.** Amazon review dataset[5] (Ni et al., 2019) includes reviews, product metadata, and links. We first combine events if a user submits multiple reviews in the same category on the same day. Then, we select data spanning from January 1, 2018 to June 30, 2018. From this dataset, we focus on users who wrote between 40 and 200 category reviews and select the top 17 categories reviewed by these users. We combine all other categories into a single "Other" category. Finally, we obtain 2,245 sequences with 127,054 events across 18 category types.

### A.3 EVENT TYPES

In this subsection, we provide a detailed mapping of event types to their corresponding textual descriptions for each dataset in Table 10-14. These event types represent the various categories of events modeled in our experiments, allowing the model to capture diverse patterns across different domains.

## B PROMPT DETAILS

In this section, we provide the detailed prompts designed for each dataset, illustrating how the event sequences are structured based on whether the event type or event time appears first. Table 15 outlines the sequence descriptions and event formatting for each dataset.

## C BASELINE DETAILS

**Neural Hawkes Process** (**NHP**) (Mei & Eisner, 2017) is a generative model that uses a continuous-time LSTM to dynamically adjust the intensity of multiple event types based on the sequence of past events, enabling accurate predictions of future event types and timings. **Self-Attentive Hawkes Process** (**SAHP**) (Zhang et al., 2020) enhances Hawkes processes by using self-attention to model event dynamics, incorporating time intervals into positional encoding, and improving predictive accuracy

---

[3] https://www.andresmh.com/nyctaxitrips/
[4] https://earthquake.usgs.gov/earthquakes/search/
[5] https://nijianmo.github.io/amazon/

Table 10: Event IDs and corresponding event types for Stack Overflow dataset.

| ID | Event Type | ID | Event Type |
|---|---|---|---|
| 0 | Yearling | 13 | Favorite Question |
| 1 | Necromancer | 14 | Populist |
| 2 | Enlightened | 15 | Announcer |
| 3 | Guru | 16 | Booster |
| 4 | Nice Question | 17 | Publicist |
| 5 | Good Question | 18 | Revival |
| 6 | Great Question | 19 | Caucus |
| 7 | Nice Answer | 20 | Constituent |
| 8 | Good Answer | 21 | Custodian |
| 9 | Great Answer | 22 | Steward |
| 10 | Popular Question | 23 | Socratic |
| 11 | Notable Question | 24 | Lifejacket |
| 12 | Famous Question | | |

Table 11: Event IDs and corresponding event types for Chicago Crime dataset.

| ID | Event Type | ID | Event Type |
|---|---|---|---|
| 0 | Theft | 10 | Burglary |
| 1 | Weapons Violation | 11 | Robbery |
| 2 | Sex Offense | 12 | Arson |
| 3 | Deceptive Practice | 13 | Offense Involving Children |
| 4 | Motor Vehicle Theft | 14 | Criminal Sexual Assault |
| 5 | Criminal Trespass | 15 | Interference With Public Officer |
| 6 | Criminal Damage | 16 | Narcotics |
| 7 | Battery | 17 | Stalking |
| 8 | Other Offense | 18 | Public Peace Violation |
| 9 | Assault | 19 | Homicide |

Table 12: Event IDs and corresponding event types for NYC Taxi Trip dataset.

| ID | Event Type | ID | Event Type |
|---|---|---|---|
| 0 | Manhattan Pickup | 4 | Bronx Dropoff |
| 1 | Manhattan Dropoff | 5 | Brooklyn Pickup |
| 2 | Queens Dropoff | 6 | Brooklyn Dropoff |
| 3 | Queens Pickup | 7 | Bronx Pickup |

Table 13: Event IDs and corresponding event types for U.S. Earthquake Dataset.

| ID | Event Type |
|---|---|
| 0 | Large |
| 1 | Medium |
| 2 | Small |

and interpretability compared to RNN-based models. **Transformer Hawkes Process** (**THP**) (Zuo et al., 2020) leverages the self-attention mechanism to efficiently capture long-term dependencies in event sequence data, improving prediction accuracy and likelihood over recurrent neural network-based models. **Attentive Hawkes Process** (**AttNHP**) (Yang et al., 2022) replaces LSTM-based architectures with attention-based models to more efficiently capture event sequences and participant embeddings, maintaining or improving prediction accuracy compared to previous neuro-symbolic and attention-based approaches. **ODE-based Temporal Point Process** (**ODETPP**) (Chen et al., 2020) leverages Neural ODEs to model temporal point processes, enabling flexible and high-fidelity representations of event sequences in continuous time by using continuous-time neural networks to condition on event history. **Meta Temporal Point Process** (**MetaTPP**) (Bae et al., 2023) introduces a meta-learning framework for TPPs, treating each event sequence as a separate task and using neu-

Table 14: Event IDs and corresponding event types for Amazon Review dataset.

| ID | Event Type | ID | Event Type |
|----|-----------|----|-----------|
| 0 | Other | 9 | Office Products |
| 1 | Tools and Home Improvement | 10 | Books |
| 2 | Pet Supplies | 11 | Home and Kitchen |
| 3 | Arts Crafts and Sewing | 12 | Sports and Outdoors |
| 4 | Electronics | 13 | Patio Lawn and Garden |
| 5 | Automotive | 14 | Movies and TV |
| 6 | Industrial and Scientific | 15 | Clothing Shoes and Jewelry |
| 7 | Grocery and Gourmet Food | 16 | Kindle Store |
| 8 | Toys and Games | 17 | Cell Phones and Accessories |

Table 15: Prompts designed for each dataset, showing how event sequences are structured with either event type first or event time first.

| Dataset | Sequence Description | Event Description | |
|---------|---------------------|-------------------|---|
| | | Event Type First | Event Time First |
| Stack Overflow | You are given a sequence of badge awards earned by a user on the Stack Overflow platform. | Each event in the sequence lists the badge name followed by the timestamp. | Each event in the sequence lists the timestamp followed by the badge name. |
| Chicago Crime | You are given a sequence of reported crime incidents that occurred in the City of Chicago. | Each event in the sequence lists the crime type followed by the timestamp. | Each event in the sequence lists the timestamp followed by the crime type. |
| NYC Taxi Trip | You are given a sequence of taxi trips taken in New York City. | Each event in the sequence lists the pick-up or drop-off location followed by the timestamp. | Each event in the sequence lists the timestamp followed by the pick-up or drop-off location. |
| U.S. Earthquake | You are given a sequence of earthquake events recorded in the United States. | Each event in the sequence lists the magnitude classification (large or small) followed by the timestamp. | Each event in the sequence lists the timestamp followed by the magnitude classification (large or small). |
| Amazon Review | You are given a sequence of product category reviews written by a user on the Amazon platform. | Each event in the sequence lists the product category followed by the timestamp. | Each event in the sequence lists the timestamp followed by the product category. |

ral processes with context sets and local history matching to learn more informative features for sequence prediction.

## D  MODEL HYPERPARAMETERS

This section details the hyperparameters used for the various models in our experiments. Table 16 summarizes the key hyperparameter configurations for each baseline (NHP, SAHP, THP, AttNHP) and our proposed model, TPP-LLM. For TPP-LLM, we include specific settings for LoRA fine-tuning, such as the rank, alpha, and dropout parameters, as well as the target attention modules.

## E  EXPERIMENTAL SETUP DETAILS

For the implementation of temporal point processes (TPPs), we used the `EasyTPP` framework (Xue et al., 2024) with the `PyTorch` back-end (Paszke et al., 2019). The large language models (LLMs) were implemented using Hugging Face's `Transformers` library (Wolf et al., 2020), and we applied parameter-efficient fine-tuning through the `PEFT` library (Mangrulkar et al., 2022).

Table 16: Hyperparameter configurations used for various models in the experiments. The model structure parameters of TPP-LLM depend on the foundation model (TinyLlama-1.1B in this table).

| Hyperparameter | NHP | SAHP | THP | AttNHP | ODETPP | MetaTPP | TPP-LLM |
|---|---|---|---|---|---|---|---|
| hidden_size | 64 | 64 | 64 | 64 | 64 | 64 | 2048 |
| time_emb_size | 16 | 16 | 16 | 16 | 16 | 64 | 2048 |
| num_layers | 2 | 2 | 2 | 2 | 2 | 2 | 22 |
| num_heads | - | 2 | 2 | 2 | - | 1 | 32 |
| batch_size | 128 | 128 | 128 | 8 | 128 | 128 | 8 |
| max_epoch | 10 | 20 | 20 | 8 | 100 | 200 | 20 |
| learning_rate | 1e-3 | 1e-3 | 1e-3 | 1e-3 | 2e-4 | 1e-4 | 5e-4 |
| num_integrals | 20 | 20 | 20 | 20 | 20 | 100 | 20 |
| lora_rank | - | - | - | - | - | - | 16 |
| lora_alpha | - | - | - | - | - | - | 16 |
| lora_dropout | - | - | - | - | - | - | 0.05 |
| target_modules | - | - | - | - | - | - | QKVO |
| beta_type | - | - | - | - | - | - | 1 |
| beta_time | - | - | - | - | - | - | 1 |

To enhance computational efficiency, we employed 4-bit quantization from the `bitsandbytes` library (Dettmers & Zettlemoyer, 2023).

# F    MORE ABLATION STUDIES

This section presents additional ablation studies, covering data variations, perturbations, intensity functions, event type formats, and prompt configurations.

## F.1    DATA VARIATIONS

We construct two additional variants of the Stack Overflow dataset to evaluate the model's performance under different data configurations, as shown in Table 17. The longer variant includes 4 years of data (2020-2023) and focuses on users with higher activity levels, specifically those earning 100–200 badges. This results in significantly longer average sequence lengths, increasing from 56 to 132. The larger variant, in contrast, spans 3 years (2021–2023) and selects users with 40–100 badges, increasing the number of sequences from 3,336 to 8,065 and introducing two additional badge types.

Table 17: Statistics overview for different variants of the Stack Overflow dataset.

| Dataset | # of Types | # of Events | # of Seq. | Avg. Seq. Length | Time Unit |
|---|---|---|---|---|---|
| Stack Overflow (Original) | 25 | 187,836 | 3,336 | 56.31 | Month |
| Stack Overflow (Longer) | 25 | 282,008 | 2,131 | 132.34 | Month |
| Stack Overflow (Larger) | 27 | 458,917 | 8,065 | 56.90 | Month |

As shown in Table 18, we evaluate the performance of various models, including two bassline (THP and AttNHP) and our model TPP-Llama, across different variants of the Stack Overflow dataset, which vary in size and sequence length. Despite longer average sequence lengths and larger numbers of sequences in the longer and larger variants, TPP-Llama consistently outperforms the other baseline models in terms of log-likelihood, accuracy, and RMSE. These results demonstrate that TPP-Llama maintains superior performance even as the dataset becomes larger and the average sequence length increases, highlighting the model's robustness and its ability to effectively handle both larger volumes of data and longer event sequences.

## F.2    DATA PERTURBATIONS

We compare the performance of the model across different perturbation ratios for the StackOverflow and Earthquake datasets. The purpose of this comparison is to evaluate the robustness of the

Table 18: Performance comparison of log-likelihood, accuracy, and RMSE across different variants of the Stack Overflow dataset.

| Model | StackOverflow-Original | StackOverflow-Longer | StackOverflow-Larger |
|---|---|---|---|
| THP | -1.877/43.81%/0.629 | -1.861/41.18%/0.561 | -2.354/40.74%/0.926 |
| AttNHP | -1.798/39.12%/0.581 | -1.820/38.45%/0.524 | -2.386/38.00%/0.803 |
| TPP-Llama | **-1.777/44.20%/0.477** | **-1.699/42.23%/0.432** | **-2.223/42.00%/0.694** |

model under specific levels of dataset perturbation (1%, 5%, and 10%) and to simulate real-world scenarios where data may be noisy or imperfect. The perturbation dataset is generated by applying a random perturbation to the original event times, where a perturbation value is drawn uniformly from the range $[-1, 1]$ and scaled by the specified perturbation ratio and each time interval. The perturbed time is then calculated by adding the perturbation to the original event time, ensuring that the perturbed time is not earlier than the previous perturbed time.

As shown in Table 19, some metrics of TPP-Llama improve with small levels of perturbation, likely due to data augmentation, which helps reduce overfitting and improves generalization. As the perturbation ratio increases further, however, performance begins to degrade, with slight increases in RMSE and minor changes in log-likelihood and accuracy, reflecting the model capability of handling noise in the data. Despite these fluctuations, TPP-Llama consistently demonstrates stable performance and overforms other baselines. These results highlight the model's resilience and ability to maintain high performance even in the presence of data perturbations.

Table 19: Performance comparison of log-likelihood, accuracy, and RMSE across different dataset perturbations.

| Model | Perturbation Ratio | StackOverflow | Earthquake |
|---|---|---|---|
| TPP-Llama | None | -1.777/44.20%/**0.477** | -0.475/62.70%/**0.288** |
| TPP-Llama | 1% | **-1.775/44.21%**/0.498 | -0.471/63.04%/0.289 |
| TPP-Llama | 5% | -1.776/44.17%/0.494 | -0.473/**63.12%**/0.294 |
| TPP-Llama | 10% | -1.776/44.18%/0.495 | **-0.470**/63.09%/0.293 |
| THP | None | -1.877/43.81%/0.629 | -0.513/62.39%/0.377 |
| AttNHP | None | -1.798/39.12%/0.581 | -0.481/61.87%/0.386 |

## F.3 INTENSITY FUNCTIONS

We conduct an ablation study to compare three intensity functions adapted for TPP-Llama, all leveraging $\boldsymbol{h}i$ as the history vector. The modified THP intensity function (Zuo et al., 2020), softplus($\alpha_k(t - t_i) + \boldsymbol{w}k^\mathsf{T}\boldsymbol{h}i + b_k$), achieves the best overall performance, balancing flexibility and stability in capturing temporal dynamics. The RMTPP intensity function (Du et al., 2016), $\exp(\alpha_k(t - t_i) + \boldsymbol{w}k^\mathsf{T}\boldsymbol{h}i + b_k)$, performs well in event type prediction but slightly underperforms in event time prediction due to its exponential nature. The SAHP intensity function (Zhang et al., 2020), softplus($\mu_i + (\eta_i - \mu_i)\exp(-\gamma_i(t - t_i))$), where $\mu i = \text{gelu}(\boldsymbol{W}\mu\boldsymbol{h}i)$, $\eta_i = \text{gelu}(\boldsymbol{W}\eta\boldsymbol{h}i)$, and $\gamma_i = \text{gelu}(\boldsymbol{W}\gamma\boldsymbol{h}i)$, shows lower performance across most metrics. These results highlight TPP-Llama's robustness with different intensity functions while indicating that the modified THP intensity function provides the most consistent and reliable results for capturing temporal patterns effectively.

Table 20: Performance comparison of log-likelihood, accuracy, and RMSE using different intensity functions on various datasets.

| Model | Intensity Function | StackOverflow | Earthquake |
|---|---|---|---|
| TPP-Llama | THP | **-1.777**/44.20%/**0.477** | **-0.475**/62.70%/**0.288** |
| TPP-Llama | RMTPP | -1.786/**44.32%**/0.484 | -0.488/**63.10%**/0.361 |
| TPP-Llama | SAHP | -4.689/42.21%/0.486 | -0.548/62.88%/0.308 |

## F.4  EVENT TYPE FORMATS

In this ablation study, we investigate the impact of using textual descriptions versus ordinal numbers for event types. The textual input uses the event type itself as the event text, while the ordinal input replaces these descriptions with numerical identifiers. As shown in Table 21, the event time prediction remains relatively consistent between the two settings. However, the event type prediction accuracy significantly drops when ordinal numbers are used, particularly for the Stack Overflow dataset, which features more complex and diverse event types compared to the Earthquake dataset. These results underscore the importance of semantic information in textual descriptions for improving event type prediction accuracy, especially in datasets with high variability in event types.

Table 21: Performance comparison of log-likelihood, accuracy, and RMSE with different event type formats.

| Event Type Format | StackOverflow | Earthquake |
| --- | --- | --- |
| Textual | -1.777/**44.20%**/**0.477** | -0.475/**62.70%**/0.288 |
| Ordinal | **-1.772**/43.42%/0.480 | **-0.473**/62.54%/**0.284** |

## F.5  PROMPT SETTINGS

Table 22 shows that using a structured prompt (denoted as "Y") generally improves log-likelihood scores for both TinyLlama models on Stack Overflow, though omitting the prompt ("N") yields slightly better event type prediction accuracy, especially on U.S. Earthquake. RMSE results are mixed, with prompts providing a small advantage on Stack Overflow but not on U.S. Earthquake. While prompts offer modest log-likelihood gains, their impact on accuracy and RMSE is inconsistent. However, adding prompts enhances the model's flexibility, particularly for multi-task scenarios.

Table 22: Performance comparison of log-likelihood, next event type prediction accuracy, and next event time prediction RMSE across different datasets with various prompt settings.

| Foundation Model | Prompt | StackOverflow | Earthquake |
| --- | --- | --- | --- |
| TinyLlama-1.1B-Intermediate | Y | **-1.777**/44.17%/0.477 | -0.476/62.65%/0.299 |
| TinyLlama-1.1B-Intermediate | N | -1.780/**44.24%**/0.478 | **-0.471**/**62.84%**/0.291 |
| TinyLlama-1.1B-Chat-v1.0 | Y | **-1.777**/44.20%/0.477 | -0.475/62.70%/**0.288** |
| TinyLlama-1.1B-Chat-v1.0 | N | -1.781/44.11%/**0.476** | -0.478/62.64%/0.304 |

