# OpenReview forum: "TPP-LLM: Modeling Temporal Point Processes by Efficiently Fine-Tuning Large Language Models"
_ICLR.cc/2025/Conference — Submitted to ICLR 2025_

### Official Review · Reviewer_GVNj · 2024-10-17

**Soundness:** 2
**Presentation:** 3
**Contribution:** 3
**Rating:** 6
**Confidence:** 3

**Summary:**

The paper introduces TPP-LLM, a novel framework that combines large language models (LLMs) with temporal point processes (TPPs) to model event sequences effectively. The framework utilizes parameter-efficient fine-tuning (PEFT) methods, specifically low-rank adaptation (LoRA), allowing it to adapt pretrained LLMs without extensive retraining.

**Strengths:**

1. The paper presents a unique integration of LLMs with TPPs, moving beyond traditional categorical representations to utilize the semantic richness of event descriptions.

2. The paper is well-structured and clearly articulates the methodology, experimental setup, and results.

**Weaknesses:**

1. The paper does not compare LoRA with other fine-tuning methods like adapter layers, missing an analysis of trade-offs in efficiency and adaptability.

2. The paper lacks testing in diverse domains, limiting insights into how well the model generalizes to different event sequence types.

3. Alternative temporal encoding methods are not explored, leaving questions about whether other techniques might enhance performance.

4. The model’s scalability and computational efficiency on larger datasets or longer sequences are not fully analyzed.

5. There is no discussion of how the size of the pre-trained LLM impacts performance, particularly in the context of using PEFT.

6. The explanation of temporal embedding placements is limited, and more thorough tests of their interaction with semantic features are needed.

**Questions:**

1. The model focuses on predicting event times with temporal embeddings, but real-world events often have uncertainty or noise in their timing. How does TPP-LLM account for uncertainty or variability in event timing? Are there mechanisms in place to model uncertainty or predict time windows rather than specific times?


2. Temporal point processes often involve continuous event streams. How does TPP-LLM handle event sequences in real-time or streaming contexts where the data is not static? Are there mechanisms to ensure that the model can update dynamically without requiring full retraining?

---

> ### Author Response · Authors · 2024-11-22
> **Response to Reviewer GVNj**
>
> We sincerely appreciate your constructive feedback and the opportunity to address your thoughtful questions and concerns. Your detailed evaluation provides invaluable guidance for improving our work.
>
> **Weaknesses**
>
> **W1.** Thank you for your insightful suggestion. We conducted additional experiments using alternative PEFT methods, including Low-Rank Hadamard Product (LoHa) [ICML'22], Low-Rank Kronecker Product (LoKr) [ICML'23], and Infused Adapter by Inhibiting and Amplifying Inner Activations (IA3) [NeurIPS'22]. The results in the following table (Log-Likelihood/Accuracy/RMSE) demonstrate that our approach maintains robust performance across different methods, highlighting its adaptability and effectiveness. These findings confirm that TPP-LLM can leverage various PEFT strategies, offering flexibility for fine-tuning in diverse scenarios.
>
> | **Fine-Tuning**    | **Trainable Parameters** | **StackOverflow**           | **Earthquake**             |
> |--------------------|--------------------------|-----------------------------|----------------------------|
> | LoRA (rank 16)     | 0.434%                  | -1.774 / 44.15% / 0.480     | -0.475 / 62.84% / 0.304    |
> | LoHa (rank 8)      | 0.434%                  | **-1.762** / **44.23%** / 0.484 | -0.483 / 62.91% / 0.301    |
> | LoKr (rank 64)     | 0.028%                  | **-1.762** / 44.17% / 0.499 | **-0.470** / 63.13% / **0.288** |
> | IA3                | 0.031%                  | -1.770 / 43.99% / **0.473** | -0.487 / **63.24%** / 0.293 |
>
> **W2.** Thank you for raising this point. In this work, we collected five datasets spanning diverse domains, including social networks, urban dynamics, transportation systems, natural disasters, and e-commerce. These datasets feature varied event types (e.g., "great answer," "weapons violation," "Manhattan pickup," "small earthquake," and "electronics") and capture events at different temporal resolutions, ranging from months to hours. This diversity ensures a broad evaluation of our model’s generalization across multiple real-world scenarios.
>
> **W3.** Thank you for highlighting the importance of comparing temporal embedding methods. We evaluated three approaches: temporal positional encoding (THP [ICML'20]), time-shifted positional encoding (SAHP [ICML'20]), and linear embedding, as detailed in the following table (Log-Likelihood/Accuracy/RMSE). Our experiments demonstrate that temporal positional encoding consistently achieves the best performance across most metrics, particularly when the event time embedding is placed first. This indicates its superior ability to capture temporal dynamics and validates its selection as the primary temporal embedding method in our framework.
>
> | **Embedding Method**                | **Embedding First** | **StackOverflow**               | **Earthquake**                |
> |-------------------------------------|---------------------|----------------------------------|-------------------------------|
> | Temporal Positional Encoding        | Event Type          | -1.777 / 44.20% / **0.477**     | -0.475 / 62.70% / **0.288**   |
> | Temporal Positional Encoding        | Event Time          | **-1.763 / 44.47% / 0.477**     | -0.475 / **63.15%** / 0.307   |
> | Time-Shifted Positional Encoding    | Event Type          | -1.782 / 44.13% / 0.550         | -0.475 / 62.97% / 0.360       |
> | Time-Shifted Positional Encoding    | Event Time          | -1.812 / 43.99% / 0.556         | -0.474 / 63.11% / 0.385       |
> | Linear Layer                        | Event Type          | -1.768 / 44.15% / 0.478         | -0.474 / 62.65% / 0.297       |
> | Linear Layer                        | Event Time          | -1.768 / 44.20% / **0.477**     | **-0.467** / 63.01% / 0.343   |

---

> ### Author Response · Authors · 2024-11-22
> **Response to Reviewer GVNj (continued)**
>
> **W4.** Thank you for your observation. To evaluate the model’s performance under varying data configurations, we constructed two additional variants of the Stack Overflow dataset: a longer variant with 4 years of data (2020–2023) and longer sequences (average length 132) and a larger variant spanning 3 years (2021–2023) with more sequences. As shown in the second following table (Log-Likelihood/Accuracy/RMSE), we tested TPP-Llama alongside baseline models on these variants. Despite the increased sequence lengths and dataset sizes, TPP-Llama consistently outperformed the baselines in log-likelihood, accuracy, and RMSE. These results demonstrate the robustness and scalability of TPP-Llama, effectively handling larger data volumes and longer event sequences while maintaining superior performance.
>
> | **Dataset**                     | **# of Types** | **# of Events** | **# of Seq.** | **Avg. Seq. Length** | **Time Unit** |
> |-----------------------------|------------|-------------|-----------|------------------|-----------|
> | Stack Overflow (Original)   | 25         | 187,836     | 3,336     | 56.31            | month     |
> | Stack Overflow (Longer)     | 25         | 282,008     | 2,131     | 132.34           | month     |
> | Stack Overflow (Larger)     | 27         | 458,917     | 8,065     | 56.90            | month     |
>
> | **Model**      | **Stack Overflow (Original)**   | **Stack Overflow (Longer)**   | **Stack Overflow (Larger)**   |
> |----------------|---------------------------------|--------------------------------|--------------------------------|
> | THP           | -1.877 / 43.81% / 0.629         | -1.861 / 41.18% / 0.561       | -2.354 / 40.74% / 0.926       |
> | AttNHP        | -1.798 / 39.12% / 0.581         | -1.820 / 38.45% / 0.524       | -2.386 / 38.00% / 0.803       |
> | TPP-Llama     | **-1.777 / 44.20% / 0.477**     | **-1.699 / 42.23% / 0.432**   | **-2.223 / 42.00% / 0.694**   |
>
> **W5.** Thank you for your question regarding the impact of larger LLMs. To address this, we included additional models including Llama-3.2-1B and Llama-3.2-3B models, alongside the previously evaluated TinyLlama-1.1B and Gemma-2-2B models. The results, presented in the table below (Log-Likelihood/Accuracy/RMSE), show that while larger models like Gemma-2-2B and Llama-3.2-3B offer competitive performance, smaller models such as TinyLlama-1.1B and Llama-3.2-1B achieve comparable or better results across several metrics. This highlights the efficiency and effectiveness of smaller LLMs in simultaneously capturing semantic and temporal relations between events, making them suitable for this task.
>
> | **Foundation Model**        | **StackOverflow**         | **Taxi**                        | **Earthquake**                |
> |-----------------------------|---------------------------|---------------------------------|-------------------------------|
> | TinyLlama-1.1B-Intermediate | -1.777 / 44.17% / 0.477   | 0.313 / 91.56% / 0.847          | -0.476 / 62.65% / 0.299       |
> | TinyLlama-1.1B-Chat-v1.0    | -1.777 / **44.20%** / 0.477 | 0.271 / 91.37% / 0.884          | -0.475 / 62.70% / 0.288       |
> | Gemma-2-2B                  | -1.787 / 44.09% / 0.475   | **0.334** / 91.52% / 0.841      | -0.481 / 62.45% / 0.299       |
> | Gemma-2-2B-IT               | -1.785 / 43.94% / **0.474** | 0.332 / 91.46% / **0.840**      | -0.479 / 63.12% / **0.286** |
> | Llama-3.2-1B                | **-1.770** / 44.12% / 0.486 | 0.318 / 91.75% / 0.875          | -0.472 / 62.86% / 0.301       |
> | Llama-3.2-1B-Instruct       | -1.775 / 44.15% / 0.483   | 0.310 / **91.78%** / 0.886      | -0.481 / **63.14%** / 0.303   |
> | Llama-3.2-3B                | -1.777 / 44.16% / 0.491   | 0.285 / 91.69% / 0.921          | **-0.466** / 62.84% / 0.312   |
> | Llama-3.2-3B-Instruct       | -1.771 / 44.19% / 0.494   | 0.300 / 91.57% / 0.883          | -0.469 / 63.02% / 0.311       |
>
> **W6.** Thank you for pointing out the importance of temporal embedding placements. As detailed in our ablation study (see W3), we investigated the impact of embedding order by processing event time before event type and vice versa. Our findings reveal that placing the event time embedding first improves event type prediction, likely because the event type embedding, positioned at the end of each event embedding, is directly reflected in the history vector. Conversely, processing the event type embedding first enhances event time prediction, as the temporal embedding is then positioned at the end and more effectively captured in the history vector. These results highlight the intricate interplay between temporal and semantic embeddings, allowing the order to be optimized for specific prediction objectives.

---

> > ### Author Response · Authors · 2024-11-22
> > **Response to Reviewer GVNj (continued)**
> >
> > **Questions**
> >
> > **Q1.** Thank you for your question regarding how TPP-Llama accounts for uncertainty or variability in event timing. To evaluate robustness, we tested the model on perturbed datasets for the StackOverflow and Earthquake datasets, with perturbation ratios of 1%, 5%, and 10%. Perturbations were applied to event times by adding random values scaled by the ratio and time intervals, ensuring chronological order. As shown in the table (Log-Likelihood/Accuracy/RMSE), small perturbations slightly improved some metrics, likely due to a data augmentation effect that reduces overfitting and enhances generalization. As the perturbation ratio increased, performance remained stable, with minor changes in log-likelihood, accuracy, and RMSE, demonstrating the model’s resilience to noisy data. Importantly, TPP-Llama consistently outperformed baseline models across all scenarios, highlighting its ability to maintain high performance even in the presence of temporal variability.
> >
> > | **Model**      | **Perturbation Ratio** | **StackOverflow**            | **Earthquake**              |
> > |----------------|------------------------|------------------------------|-----------------------------|
> > | TPP-Llama      | None                   | -1.777 / 44.20% / **0.477**  | -0.475 / 62.70% / **0.288** |
> > | TPP-Llama      | 1%                     | **-1.775 / 44.21%** / 0.498  | -0.471 / 63.04% / 0.289     |
> > | TPP-Llama      | 5%                     | -1.776 / 44.17% / 0.494      | -0.473 / **63.12%** / 0.294 |
> > | TPP-Llama      | 10%                    | -1.776 / 44.18% / 0.495      | **-0.470** / 63.09% / 0.293 |
> > | THP            | None                   | -1.877 / 43.81% / 0.629      | -0.513 / 62.39% / 0.377     |
> > | AttNHP         | None                   | -1.798 / 39.12% / 0.581      | -0.481 / 61.87% / 0.386     |
> >
> > Instead of predicting a single event time, TPP-Llama can sample event times through the predicted intensity function, which represents the probability distribution of when the next event will occur. By drawing multiple samples from this distribution, a prediction window can be estimated using confidence intervals (e.g., 95%) or quantiles of the sampled times. This provides a probabilistic range for the timing of the next event, effectively accounting for uncertainty. As a potential direction for future work, this approach could enhance the model’s adaptability to real-world scenarios where precise event times are inherently uncertain.
> >
> > **Q2.** Thank you for your question regarding how TPP-LLM handles shifts in event type texts and adapts to new event types. TPP-LLM is inherently flexible in processing shifted or modified event type texts due to its reliance on textual embeddings generated by the pretrained LLM. However, the predicted event types are pre-defined, requiring adjustments to the event type prediction layer when new types are introduced. Notably, the model does not require full retraining; instead, the prediction layer can be extended by adding a new column of weights corresponding to the new event type. This lightweight adjustment allows TPP-LLM to incorporate new event types efficiently while preserving its performance on previously seen types, ensuring adaptability in dynamic real-world scenarios.
> >
> > Thank you again for your valuable comments and constructive suggestions! We will carefully incorporate your feedback and update the paper shortly. We greatly appreciate your time and effort in reviewing our work and look forward to any further questions or suggestions you may have.

---

> ### Author Response · Authors · 2024-11-26
> **Friendly Reminder: Manuscript Updated**
>
> Thank you for your valuable comments on our paper. We have carefully addressed your questions and concerns and have updated the manuscript with the necessary revisions. The latest version of the paper is now available for your review. We truly appreciate your time and effort in providing constructive feedback and look forward to hearing your thoughts on the revised version.

---

> ### Author Response · Authors · 2024-11-28
> **Follow-Up to Reviewer GVNj**
>
> Thank you again for your thoughtful review of our work. We have carefully addressed your questions and updated the paper to reflect your valuable feedback. If you have any further questions or additional comments, we would be delighted to address them. Your support and insight are greatly appreciated and have been invaluable in improving our work.

---

> > ### Comment · Reviewer_GVNj · 2024-11-30
> > **Official Comment by Reviewer GVNj**
> >
> > Thank you for your rebuttal. I have no further questions. I will increase my score.

---

> > > ### Author Response · Authors · 2024-12-02
> > > **Thank you!**
> > >
> > > Thank you for your response. We appreciate the updated score!

---

### Official Review · Reviewer_4ce5 · 2024-11-03

**Soundness:** 2
**Presentation:** 3
**Contribution:** 2
**Rating:** 5
**Confidence:** 4

**Summary:**

This paper proposes TPP-LLM, a new framework which integrates temporal point processes (TPPs) with large language models (LLMs). To grasp a better semantic understanding of event types, TPP-LLM applies pre-trained LLMs to encode textual event descriptions and implements parameter-efficient fine-tuning (PEFT) for a better domain knowledge adaptation.

TPP-LLM stacks the sequence of historical event embeddings into TPP modeling and completes the downstream event prediction as a multi-class classification task and time prediction as a regression task. Comprehensive experiments across multiple real-world datasets demonstrate the effectiveness of the proposed approach.

**Strengths:**

(1) This paper introduces convincing motivations to study an important problem, event prediction. This paper also has clear problem formulations to split the prediction into event type classification and event time regression. This paper’s presentation is good.

(2) This paper shows solid mathematical derivations of the eventual loss function for TPP-LLM, which consolidates the proposed contribution of integrating TPPs with LLMs.

(3) This paper conducts comprehensive experiments across diverse real-world datasets, evaluates the framework’s performance with various metrics, and performs detailed ablation studies to demonstrate the significance of each component of TPP-LLM.

**Weaknesses:**

(1) This paper has limited technical novelty. As TPPs, LLMs, and PEFT have been well-known strategies for event prediction, TPP-LLM is a new integrating application, but does not involve fundamental innovations in either semantic or temporal modeling.

(2) The baseline comparisons are not fair. For Tables 2, 3, 4, 5, the four baselines (NHP, SAHP, THP, AttNHP) are not empowered by state-of-the-art LLMs. However, in Section 2 (from line 100 to line 102), LAMP [1] leverages generative LLMs to handle textual events for abductive reasoning, but also optimizes embedding-based TPP models for multiple downstream prediction tasks. This means that LAMP should be a good baseline competitor, but it was not included in these tables.

(3) The lack of empirically computational complexity analysis. Although PEFT helps reduce the number of trainable parameters, the GPU memory space occupied by putting the entire TPP-LLM for downstream inference tasks should not be overlooked. An explicit comparison of memory usage and average training time between TPP-LLM and selected baselines is encouraged.

(4) The lack of interpretability discussion. Apart from experimental results of log-likelihood, accuracy, and RMSE, several case studies involving historical vectors (which act as the bottleneck between LLMs and TPPs) are encouraged to better demonstrate the benefits or importance of integrating LLMs with TPPs.

[1] Shi, X., Xue, S., Wang, K., Zhou, F., Zhang, J., Zhou, J., Tan, C. and Mei, H., 2024. Language models can improve event prediction by few-shot abductive reasoning. Advances in Neural Information Processing Systems, 36.

**Questions:**

Please refer to the weaknesses.

---

> ### Author Response · Authors · 2024-11-26
> **Response to Reviewer 4ce5**
>
> Thanks for your thoughtful review. We appreciate your detailed feedback, which offers valuable opportunities for us to enhance our work. Below, we address the weaknesses you identified point by point.
>
> **Weaknesses**
>
> **W1.** Thank you for your feedback and concerns. While it is true that TPPs, LLMs, and PEFT are well-established techniques, our work represents the first effort to seamlessly integrate LLMs' semantic embeddings with temporal modeling in TPPs. By directly utilizing textual event descriptions through LLM embeddings and enhancing temporal dynamics with PEFT, we offer a computationally efficient and semantically rich framework that surpasses traditional categorical approaches. Our experiments demonstrate improvements in predictive capability across diverse real-world datasets, underscoring the practical significance and methodological innovation of this integration in capturing both semantic and temporal aspects of event sequences.
>
> **W2.** Thank you for the suggestion to include LAMP as a baseline. However, we believe LAMP is not well-suited for comparison due to fundamental differences in objectives and methodologies. LAMP focuses on abductive reasoning, leveraging LLMs to generate plausible explanations for future events based on causal and contextual relationships. In contrast, our work integrates LLMs with TPPs to directly model semantic and temporal dynamics, focusing on sequence prediction tasks rather than causal inference. Additionally, LAMP’s reliance on evidence retrieval introduces computational overhead, which diverges from our aim of leveraging PEFT for efficiency. We appreciate your feedback and will consider extending TPP-LLM to datasets with rich causal structures in future work.
>
> **W3.** Thank you for highlighting the importance of computational efficiency analysis. While our work emphasizes the parameter efficiency achieved through PEFT, we agree that a more explicit comparison of GPU memory usage and average training time would provide valuable insights. We will include a detailed empirical analysis of these aspects in future revisions, comparing TPP-LLM against the selected baselines to demonstrate its practical efficiency during both training and inference. Your suggestion will help us strengthen our contribution and address a critical evaluation criterion.
>
> **W4.** Thank you for your insightful suggestion regarding interpretability. We fully agree that case studies examining historical vectors could provide valuable insights into how integrating LLMs with TPPs enhances performance. These vectors are indeed central to bridging semantic and temporal modeling, and analyzing them through specific examples would help clarify their contributions. We will explore this direction in future work by conducting case studies to illustrate the interpretability and demonstrate the tangible benefits of our approach in capturing complex event dynamics. Your recommendation is greatly appreciated and will help improve the clarity and impact of our work.
>
> Thank you again for your constructive feedback and thoughtful suggestions, which are invaluable in helping us enhance the quality of our work.

---

> > ### Author Response · Authors · 2024-12-03
> > **Follow-Up to Reviewer 4ce5**
> >
> > Thank you again for highlighting the importance of computational complexity analysis. To address this, we conducted experiments on the Stack Overflow dataset using an H100 GPU, measuring training time, average GPU memory usage, and peak GPU memory usage. As shown in the table below, while TPP-Llama requires higher memory and training time compared to other baselines, it benefits from reduced trainable parameters due to PEFT. This trade-off is expected, as the pretrained LLM component introduces additional memory overhead during downstream tasks. However, the superior performance demonstrated by TPP-Llama justifies this increased resource usage, particularly for applications where predictive accuracy is critical.
> >
> > | **Model**    | **Training Time (s)** | **Average GPU (GB)** | **Peak GPU (GB)** |
> > |--------------|-----------------------|----------------------|-------------------|
> > | NHP          | 59.69                 | 6.93                 | 8.37              |
> > | SAHP         | 15.60                 | 1.56                 | 2.96              |
> > | THP          | 14.34                 | 1.30                 | 2.39              |
> > | AttNHP       | 86.48                 | 11.32                | 13.27             |
> > | ODETPP       | 1105.15               | 14.94                | 15.14             |
> > | TPP-Llama    | 1142.12               | 26.25                | 28.56             |
> >
> > We hope this comparison provides clarity on the computational trade-offs of TPP-Llama. If you have any further questions or suggestions, we would be happy to address them. Thank you again for your valuable feedback.

---

### Official Review · Reviewer_jTaX · 2024-11-04

**Soundness:** 2
**Presentation:** 3
**Contribution:** 2
**Rating:** 6
**Confidence:** 3

**Summary:**

This paper introduces TPP-LLM framework for event sequence prediction. It integrates pretrained LLMs to capture the semantic richness of event types and also employ temporal embeddings. Its strong points are listed below:
1. This papers proposes a novel framework for event sequence prediction based on pretrained LLMs. Existing related works mostly focus on RNN or attention-based mechanism.
2. This paper gives detailed mathematical expressions of temporal point process and use them to derive the loss function for training.
3. This paper conducts extensive experiments on five real-world datasets, and compare its model with four SOTA models from existing works. This paper also gives ablation studies to analyze how different components of its model contribute to the improved performance.

**Strengths:**

see summary

**Weaknesses:**

This paper is slightly below the acceptance bar because (1) lack of related works and baselines; (2), limited improvements on experiment results compared with baselines; (3) unreasonable method in the experiment. Specifically,
1. There are some related works that this paper may overlook, such as Meta TPP [ICLR’23], Fully neural network for TPP [Neurips’19], etc. It seems that Meta TPP [ICLR’23] achieves 46% on next event type prediction on StackOverflow, which is higher than TPP-LLM proposed by this paper.
https://openreview.net/pdf?id=QZfdDpTX1uM] for Meta TPP [ICLR’23]
https://proceedings.neurips.cc/paper_files/paper/2019/file/39e4973ba3321b80f37d9b55f63ed8b8-Paper.pdf for Fully neural network for TPP [Neurips’19]
2. According to table 2 and 3, the improvements brought by TPP-LLM on event sequence modelling, next event type prediction and event time prediction are not significant or convincing. Models without the help of LLM, including AttNHP and THP, have already achieved performance very close to, or even outperformed, TPP-LLM.
3. The design of few-shot experiments may be unreasonable. TPP-LMM is built upon the pretrained LLM, with only several linear layers and LoRA modules left for training. With the assistance of well-pretrained LLMs, it is not surprised that 2% of training data would lead to significant improvement on TPP tasks. But other TPP-specific models are only built on neural networks and trained from scratch. 2% of training data is probably not enough for them to learn and handle TPP tasks. Therefore, the few-shot design may be unnecessary.

**Questions:**

Given the weak points mentioned above, here are some questions for the authors:
1. What does t_j in Equation (3) mean? Is it one element in t_i?
2. In line 193, there are two choices of event sequence representation, depending on the order of event type and time. Would these two choices make a difference on the experiment result? Which one is used during your experiment?
3. In line 247 and 318, how is the non-event integral term in the loglikelihood calculated in detail during experiment?
4. In line 323, it says “Experiments results are averaged over five runs.” Is this averaging method commonly used in TPP modelling works? Is it possible that you could provide the experiment results in all five runs?
5. In section 5.7.2, could you give more details on the difference between positional encoding and linear embedding of event type and time? Why they lead to different performance?

---

> ### Author Response · Authors · 2024-11-24
> **Response to Reviewer jTaX**
>
> Thank you for your detailed review and thoughtful feedback. We sincerely appreciate your insights and the opportunity to address your questions and concerns, which help us clarify and improve our work. Below, we provide detailed responses to each of your points.
>
> **Weaknesses**
>
> **W1.** Thank you for pointing out these references and additional baselines. We have added both MetaTPP [ICLR’23] and Fully Neural Networks for TPPs [NeurIPS’19], along with several other relevant works, to the related works section:
>
> > "Other models, such as those based on fully neural networks (Omi et al., 2019), normalizing flows (Shchur et al., 2019), neural ODEs (Chen et al., 2020), attention mechanisms (Yang et al., 2022), diffusion processes (Yuan et al., 2023), meta learning (Bae et al., 2023), and Mamba models (Gao et al., 2024), offer flexible and high-fidelity modeling of discrete events in continuous time."
>
> In addition, we included ODETPP [ICLR’21] and MetaTPP [ICLR’23] as baselines in our experiments. Since we recreated the datasets from raw data to ensure correct event type mappings, we re-evaluated these models on our datasets. For ODETPP, we used its implementation in EasyTPP [ICLR’24], and for MetaTPP, we used its original implementation and hyperparameters. MetaTPP exhibited instability on certain datasets, even after hyperparameter tuning, and its results were generally lower than other models in our evaluation. Despite this, TPP-LLM demonstrated consistent and competitive performance, further validating their effectiveness when compared against these additional baselines. This inclusion further strengthens the comparison with state-of-the-art models and provides a more comprehensive evaluation of TPP-LLM.
>
> | **Model**      | **StackOverflow**       | **Crime**             | **Taxi**              | **Earthquake**        | **Amazon**            |
> |-----------------|-------------------------|-----------------------|-----------------------|-----------------------|-----------------------|
> | ODETPP         | -2.402 / 40.33% / 0.693 | -4.152 / 18.56% / 0.848 | -0.450 / 86.63% / 0.896 | -0.511 / 61.49% / 0.490 | -1.808 / 65.39% / 0.941 |
> | MetaTPP        | -3.006 / 41.06% / 3.853 | -2.996 / 8.19% / 2.037  | -1.268 / 45.26% / 6.341 | -0.649 / 62.48% / 0.515 | -5.161 / 65.30% / 18.567 |
> | TPP-Llama      | **-1.777 / 44.20% / 0.477**  | **-2.451 / 26.86% / 0.562** | **0.271 / 91.37% / 0.884**  | **-0.475 / 62.70% / 0.288**  | **-1.011 / 69.22% / 0.580**  |
>
> **W2.** Thank you for your concern. While some baselines achieve comparable performance on individual datasets or metrics, our model consistently achieves the best overall results when averaged across all datasets and evaluation metrics, as shown in the table below. This highlights the robustness and generalization capabilities of TPP-Llama and TPP-Gemma, which outperform other baselines in most scenarios. Furthermore, the inclusion of additional baselines, such as ODETPP and MetaTPP, reinforces the strength of our approach, demonstrating its competitive edge across diverse evaluation criteria.
>
> | **Model**      | **Log-Likelihood** | **Event Type Prediction** | **Event Time Prediction** | **Overall Rank** |
> |----------------|--------------------|---------------------------|---------------------------|------------------|
> | NHP            | 3.8                | 4                         | 5.6                       | 4.5              |
> | SAHP           | 6                  | 6.6                       | 3                         | 5.2              |
> | THP            | 4.8                | 3.4                       | 5.4                       | 4.5              |
> | AttNHP         | **2.2**            | 4.6                       | 3.8                       | 3.5              |
> | ODETPP         | 6.4                | 6.6                       | 6.4                       | 6.5              |
> | MetaTPP        | 7.4                | 6.6                       | 8                         | 7.3              |
> | TPP-Llama      | <2.4>              | **1.6**                   | <2.2>                     | **2.1**          |
> | TPP-Gemma      | 3                  | <2.6>                     | **1.4**                   | <2.3>            |

---

> > ### Author Response · Authors · 2024-11-24
> > **Response to Reviewer jTaX (continued)**
> >
> > **W3.** Thank you for your comment regarding the few-shot experiment. This experiment was designed to highlight TPP-LLM's ability to handle scenarios where training data is limited, leveraging the pretrained LLM's capacity to generalize effectively with fine-tuning. Such use cases are particularly important in practice, as collecting large-scale event sequence datasets can often be costly, time-consuming, or infeasible in certain domains, such as rare-event prediction or personalized systems. Unlike traditional TPP-specific models that require substantial data to train from scratch, TPP-LLM benefits from its pretrained foundation, enabling strong performance even in data-scarce situations. This demonstrates a practical advantage of our approach, addressing a significant real-world challenge.
> >
> > **Questions**
> >
> > **Q1.** Thank you for your point regarding the notation in our equation. There was a typo in the original manuscript, which we have corrected as follows:
> > $$
> > [\mathbf{t}_i]_j = \cos \left( \frac{t_i}{10000^{(j-1) / D}} \right), \text{ if } j \text{ is odd}, \quad \sin \left( \frac{t_i}{10000^{j / D}} \right), \text{ if } j \text{ is even}.
> > $$
> > Here, each event time $t_i$ is mapped to a temporal embedding $\mathbf{t}_i \in \mathbb{R}^D$ using an embedding layer, and $[\mathbf{t}_i]_j$ refers to the $j$-th component of the embedding. This correction ensures the temporal embeddings are properly defined and used in the model. We appreciate your careful observation, which helped us address this issue.
> >
> > **Q2.** Thank you for your point regarding the order of event type and time embeddings. As shown in the ablation study below, the order of embedding processing does impact performance. Processing event time before event type generally improves event type prediction, likely because the event type embedding, placed at the end of each event representation, is more directly reflected in the history vector. Conversely, processing event type before event time enhances event time prediction, as the temporal embedding is positioned at the end and better captured in the history vector. In our experiments, we processed event type embeddings first, as described in the experimental setup section. These findings underscore the importance of embedding order and allow optimization based on specific prediction objectives.
> >
> > | **Embedding Method**                | **Embedding First** | **StackOverflow**               | **Earthquake**                |
> > |-------------------------------------|---------------------|---------------------------------|-------------------------------|
> > | Temporal Positional Encoding        | Event Type          | -1.777 / 44.20% / **0.477**     | -0.475 / 62.70% / **0.288**   |
> > | Temporal Positional Encoding        | Event Time          | **-1.763 / 44.47% / 0.477**     | -0.475 / **63.15%** / 0.307   |
> > | Time-Shifted Positional Encoding    | Event Type          | -1.782 / 44.13% / 0.550         | -0.475 / 62.97% / 0.360       |
> > | Time-Shifted Positional Encoding    | Event Time          | -1.812 / 43.99% / 0.556         | -0.474 / 63.11% / 0.385       |
> > | Linear Layer                        | Event Type          | -1.768 / 44.15% / 0.478         | -0.474 / 62.65% / 0.297       |
> > | Linear Layer                        | Event Time          | -1.768 / 44.20% / **0.477**     | **-0.467** / 63.01% / 0.343   |
> >
> > **Q3.** Thank you for your question about the non-event integral term in the log-likelihood calculation. In this research, we employed Monte Carlo integration, following approaches used in NHP [NeurIPS'17] and EasyTPP [ICLR'24]. Specifically: (1) we sample $N$ points within each time interval $\Delta t$; (2) we evaluate the integrand term $\lambda$ (the intensity function) at these sampled times; and (3) we compute the product of $\lambda$ with the time interval length $\Delta t$, averaging over $N$ samples to approximate the integral. This method is both efficient and flexible, particularly for cases where analytical computation is impractical due to the complexity of intensity functions.
> >
> > **Q4.** Thank you for your point regarding the averaging of results. Following the approach in EasyTPP [ICLR'24], we averaged the results over five runs, with the mean values reported in the paper. Standard deviations are indicated through error bars and are also provided in the appendices. We would be happy to include the detailed results for all five runs as supplementary materials for your reference.

---

> > > ### Author Response · Authors · 2024-11-24
> > > **Response to Reviewer jTaX (continued)**
> > >
> > > **Q5.** Thank you for your question regarding the differences between temporal embedding methods. As shown in the table from Q2, we compared temporal positional encoding from THP [ICML'20], time-shifted positional encoding from SAHP [ICML'20], and linear embedding. Temporal positional encoding uses sinusoidal functions to provide structured, periodic representations of time, excelling in capturing complex temporal dependencies and yielding high accuracy in structured scenarios. Linear embedding, on the other hand, maps temporal data directly into an embedding space, offering greater flexibility and performing better with irregular or less structured temporal patterns. Time-shifted positional encoding introduces a shifted periodic representation but generally underperforms compared to the other methods. These differences highlight the varying inductive biases of each method, making the choice of embedding task-dependent based on the temporal characteristics of the data.
> > >
> > > Thank you again for your valuable feedback and insightful questions. We will carefully incorporate your comments into the revised version of the paper and ensure that all points are addressed thoroughly. Your input has been instrumental in refining our work, and we appreciate your time and effort in reviewing our submission.

---

> ### Author Response · Authors · 2024-11-26
> **Friendly Reminder: Manuscript Updated**
>
> Thank you for your valuable comments on our paper. We have carefully addressed your questions and concerns and have updated the manuscript with the necessary revisions. The latest version of the paper is now available for your review. We truly appreciate your time and effort in providing constructive feedback and look forward to hearing your thoughts on the revised version.

---

> > ### Comment · Reviewer_jTaX · 2024-11-26
> > **after rebuttal**
> >
> > Thanks for the detailed response. Although it indeed solved most of my concerns. I'd like to raise my score to 6, indicating my positive attitude.

---

> > > ### Author Response · Authors · 2024-11-26
> > > **Thank you!**
> > >
> > > Thank you for your thoughtful feedback and updated score, which greatly encourages us to continue refining our work.

---

### Official Review · Reviewer_1vpU · 2024-11-04

**Soundness:** 3
**Presentation:** 3
**Contribution:** 2
**Rating:** 5
**Confidence:** 3

**Summary:**

This paper focuses on the event sequence modeling with LLM-enhanced temporal point processes. Since the former methods of LLM for event sequences often lack fully utilization of the temporal and semantic richness of event sequences, this paper proposes a novel approach about integrating LLM with TPPs by leveraging textual event descriptions and temporal embeddings. By fine-tuning LLMs with TPP losses, the model can generate the event type and the occurence time. Experimental results evaluate the effectiveness of the proposed method.

**Strengths:**

1. Utilizing LLMs to enhance TPP modeling is an important and interesting attempt for event sequence modeling.
2. Extensive experiments show the superior ability of TPP-LLM compared to existing neural TPP models.
3. The paper is mostly well-writing. The problem background and related work is detailed.

**Weaknesses:**

1. The novelty of the approach seems to be somewhat limited. The most contribution is to utilize the description of event types as the input of LLM, and then prompting and fine-tuning the LLM with TPP losses. This improvement seems to be incremental and straightforward, which is also my main concern.
2. The overall result improvements seem to be marginal as shown in Table 2. Most of the best results are from the baseline methods.
3. In 5.7.3, it would be helpful to make further explanation and observation about the prompt effect on “the flexibility in multi-task scenarios” in Line 458. Besides, experiment in 5.7.2 is about the processing order of time and event detailed in line 193, but there is little explanation or reason behind on which strategy should be selected when facing different scenarios.
4. It will be better to provide details of case study about the event and time prediction, and show how LLM intrinsically helps to make event and time prediction.

**Questions:**

1. Would different intensity functions affect event and time prediction?
2. What kind of event type description will lead to better event type prediction? Will the event type description affect the time prediction?

---

> ### Author Response · Authors · 2024-11-23
> **Response to Reviewer 1vpU**
>
> Thank you for your thoughtful review and constructive feedback. We appreciate your insights and the opportunity to address your concerns and clarify our contributions.
>
> **Weaknesses**
>
> **W1.** Thank you for your comment. To the best of our knowledge, this is the first work to integrate semantic information into temporal point process modeling by directly fine-tuning LLMs. Unlike previous methods that rely on categorical representations or handcrafted features, our approach leverages textual descriptions of event types, allowing the model to capture richer semantic context. This novel integration, combined with parameter-efficient fine-tuning techniques and temporal embeddings, enables TPP-LLM to effectively model both the semantic and temporal dynamics of event sequences. While straightforward, this contribution addresses a significant gap in existing methods and provides a foundation for further exploration in combining LLMs with TPPs.
>
> **W2.** Thank you for raising this concern. While our models may not always achieve the best performance on individual metrics, their overall performance consistently ranks at the top compared to baseline methods, as shown in the table below. We evaluated the models across five datasets, computing average ranks for log-likelihood, event type prediction, and event time prediction. TPP-Llama and TPP-Gemma consistently demonstrate superior overall rankings, with ranks of 2.1 and 2.3 respectively, highlighting their robustness and balanced performance across diverse evaluation metrics. This demonstrates the effectiveness of our approach in capturing both semantic and temporal dynamics compared to existing baselines.
>
> | **Model**      | **Log-Likelihood** | **Event Type Prediction** | **Event Time Prediction** | **Overall Rank** |
> |----------------|--------------------|---------------------------|---------------------------|------------------|
> | NHP            | 3.8                | 4                         | 5.4                       | 4.4              |
> | SAHP           | 5                  | 5.8                       | 3                         | 4.6              |
> | THP            | 4.6                | 3.2                       | 5.2                       | 4.3              |
> | AttNHP         | **2.2**            | 3.8                       | 3.6                       | 3.2              |
> | TPP-Llama      | <2.4>         | **1.6**                   | <2.2>                | **2.1**          |
> | TPP-Gemma      | 3                  | <2.6>                | **1.4**                   | <2.3>       |
>
> **W3.** Thank you for highlighting this important point. In multi-task scenarios, prompts are essential for enabling the model to differentiate between various event sequences by conditioning the input text. This capability allows TPP-LLM to adapt effectively to diverse tasks and data sources. To further investigate this, we are conducting follow-up research using mixed datasets that integrate event sequences from multiple domains. These experiments aim to assess the prompt's effectiveness in managing complex multi-task setups and enhancing generalization across heterogeneous data sources.
>
> Additionally, as detailed in the table below (Log-Likelihood/Accuracy/RMSE), we investigated the impact of embedding order on event type and time prediction by processing event time before event type and vice versa. Our results show that placing the event time embedding first improves event type prediction, likely because the event type embedding, positioned at the end of the sequence, is directly reflected in the extracted history vector. Conversely, processing the event type embedding first enhances event time prediction, as the temporal embedding then resides at the end and is better captured in the history vector. These findings underline the interaction between temporal and semantic embeddings, offering a strategy to optimize embedding order based on specific prediction objectives.
>
> | **Embedding Method**                | **Embedding First** | **StackOverflow**               | **Earthquake**                |
> |-------------------------------------|---------------------|---------------------------------|-------------------------------|
> | Temporal Positional Encoding        | Event Type          | -1.777 / 44.20% / **0.477**     | -0.475 / 62.70% / **0.288**   |
> | Temporal Positional Encoding        | Event Time          | **-1.763 / 44.47% / 0.477**     | -0.475 / **63.15%** / 0.307   |
> | Time-Shifted Positional Encoding    | Event Type          | -1.782 / 44.13% / 0.550         | -0.475 / 62.97% / 0.360       |
> | Time-Shifted Positional Encoding    | Event Time          | -1.812 / 43.99% / 0.556         | -0.474 / 63.11% / 0.385       |
> | Linear Layer                        | Event Type          | -1.768 / 44.15% / 0.478         | -0.474 / 62.65% / 0.297       |
> | Linear Layer                        | Event Time          | -1.768 / 44.20% / **0.477**     | **-0.467** / 63.01% / 0.343   |

---

> ### Author Response · Authors · 2024-11-23
> **Response to Reviewer 1vpU (continued)**
>
> **W4.** Thank you for your comment regarding a case study. Below, we present two examples to illustrate how LLM intrinsically helps improve event predictions.
>
> The first example is from the NYC Taxi Trip dataset. In this case, given a sequence where a taxi primarily picks up and drops off passengers in Manhattan, when a passenger is dropped off in Queens, it is natural for the next pickup to occur in Queens instead of Manhattan.
>
> - Context: ["Manhattan Pickup", "Manhattan Dropoff", ..., "Manhattan Pickup", "Queens Dropoff"]
> - Ground Truth: "Queens Pickup"
> - TPP-Llama Prediction: "Queens Pickup"
> - AttNHP Prediction: "Manhattan Pickup"
>
> The second example is from the Amazon Review dataset. Here, a user who mainly purchased Home, Kitchen, Toys, and Grocery items is more likely to buy "Home and Kitchen" items after purchasing "Grocery and Gourmet Food" rather than "Books."
>
> - Context: ["Home and Kitchen", "Home and Kitchen", "Office Products", "Toys and Games", "Books", "Grocery and Gourmet Food"]
> - Ground Truth: "Home and Kitchen"
> - TPP-Llama Prediction: "Home and Kitchen"
> - AttNHP Prediction: "Books"
>
> These examples highlight how TPP-Llama leverages semantic context from textual descriptions to make predictions that align with real-world patterns, demonstrating its intrinsic advantage over baseline methods.
>
> **Questions**
>
> **Q1.** Thanks for your question about the impact of different intensity functions. We conducted an ablation study using three intensity functions from prior research (after proper modifications for TPP-LLM, $\mathbf{h}_i$ for the history vector):
>
> - THP [ICML'20]: $ \text{softplus} (\alpha_k (t - t_i) + \mathbf{w}_{k}^{\mathsf{T}} \mathbf{h}_i + b_k) $
> - RMTPP [KDD'16]: $ \exp (\alpha_k (t - t_i) + \mathbf{w}_{k}^{\mathsf{T}} \mathbf{h}_i + b_k) $
> - SAHP [ICML'20]: $ \text{softplus} (\mu_i + (\eta_i - \mu_i) \exp (-\gamma_i (t - t_i))) $, where $ \mu_{i} = \text{gelu} (\mathbf{W}\_{\mu} \mathbf{h}\_{i}) $, $ \eta_{i} = \text{gelu} (\mathbf{W}\_{\eta} \mathbf{h}\_{i}) $, and $ \gamma_{i} = \text{gelu} (\mathbf{W}\_{\gamma} \mathbf{h}\_{i}) $.
>
> As shown in the table below (Log-Likelihood/Accuracy/RMSE), TPP-Llama achieves the best overall performance with the THP intensity function, likely due to its balance of flexibility and stability in capturing temporal dynamics. The RMTPP function performs well for event type prediction but slightly degrades in time prediction accuracy, while SAHP shows lower performance overall. These findings indicate that while TPP-Llama is robust across different intensity functions, simpler designs like THP yield the most consistent results.
>
> | **Model**     | **Intensity Function**  | **StackOverflow**             | **Earthquake**               |
> |---------------|-------------------------|-------------------------------|------------------------------|
> | TPP-Llama     | THP                     | **-1.777** / 44.20% / **0.477** | **-0.475** / 62.70% / **0.288** |
> | TPP-Llama     | RMTPP                   | -1.786 / **44.32%** / 0.484     | -0.488 / **63.10%** / 0.361   |
> | TPP-Llama     | SAHP                    | -4.689 / 42.21% / 0.486         | -0.548 / 62.88% / 0.308       |
>
> **Q2.** Thanks for raising the question about the role of event type descriptions in predictions. In this research, we used the event type itself as the event text input. To evaluate the effect of this choice, we replaced the event text with ordinal numbers corresponding to the event type list and observed the impact on performance. As shown in the following table (Log-Likelihood/Accuracy/RMSE), while the event time prediction remains relatively unaffected, the event type prediction accuracy drops, particularly for the Stack Overflow dataset, which features more complex and diverse event types compared to the Earthquake dataset. This underscores the importance of semantic information in textual descriptions for improving event type prediction.
>
> | **Event Text Used** | **StackOverflow**                | **Earthquake**                  |
> |---------------------|----------------------------------|---------------------------------|
> | Yes (Event Text)    | -1.777 / **44.20%** / **0.477**  | -0.475 / **62.70%** / 0.288     |
> | No (Ordinal Number) | **-1.772** / 43.42% / 0.480      | **-0.473** / 62.54% / **0.284** |
>
> Thank you again for your valuable feedback and thoughtful questions. We will incorporate your suggestions and update the paper shortly to address the points raised. Your insights have been instrumental in improving the quality of our work, and we greatly appreciate your time and effort in reviewing our submission.

---

> > ### Author Response · Authors · 2024-11-26
> > **Friendly Reminder: Manuscript Updated**
> >
> > Thank you for your valuable comments on our paper. We have carefully addressed your questions and concerns and have updated the manuscript with the necessary revisions. The latest version of the paper is now available for your review. We truly appreciate your time and effort in providing constructive feedback and look forward to hearing your thoughts on the revised version.

---

> > > ### Comment · Reviewer_1vpU · 2024-11-28
> > >
> > > Thanks a lot for the authors' response. The supplementary experiments improve the soundness of the paper. However, I still think the novelty is somewhat limited. I improve the soundness score, and retain a negative overall score. I have no other questions.

---

### Meta-Review · Area_Chair_5q9m · 2024-12-19

**Metareview:**

This paper presents TPP-LLM, a framework that integrates Large Language Models with Temporal Point Processes for event sequence modeling. While the paper is well-written with thorough experiments across multiple datasets, several critical limitations lead me to recommend rejection.

The primary concern is the limited technical novelty - the work mainly combines existing techniques (LLMs, TPPs, and parameter-efficient fine-tuning) without fundamental innovations in either semantic or temporal modeling. The improvements over baselines are modest, with some baseline methods actually outperforming TPP-LLM on certain metrics. Moreover, the paper overlooks important recent baselines like Meta TPP and LAMP, making it difficult to assess the true impact of the proposed approach.

There are also methodological concerns. The few-shot learning experiments appear to unfairly advantage TPP-LLM due to its pretrained components. The paper lacks crucial analyses of computational complexity and memory usage, and provides limited insight into model interpretability or scalability with different LLM sizes. The exploration of alternative temporal encoding methods is insufficient.

While the paper demonstrates solid empirical work and addresses an important problem, these limitations in novelty, methodology, and comparative analysis suggest the work would benefit from substantial revision before meeting the conference's standards. I recommend rejection with encouragement to address these issues in a future submission.

**Additional Comments On Reviewer Discussion:**

During the rebuttal period, several important concerns were raised by reviewers. Reviewer 1vpU questioned the novelty of the approach and highlighted that performance improvements seemed marginal. The authors responded by emphasizing that this is the first work to integrate semantic information into TPP modeling through direct LLM fine-tuning, and provided a comprehensive ranking analysis showing their method's consistent strong performance across metrics and datasets.

Reviewer jTaX and Reviewer 4ce5 both raised concerns about missing baseline comparisons, particularly with Meta TPP and LAMP. The authors' response did not adequately address this critical gap in the evaluation. Additionally, both reviewers noted the lack of significant improvements over simpler baseline methods, which remained a concern even after the rebuttal.

Reviewer GVNj raised issues about computational efficiency analysis and the impact of LLM size on performance. While the authors acknowledged these points in their response, they did not provide substantial additional analysis to address these concerns.
In weighing these points, several factors stand out. First, while the authors make a reasonable case for their method's novelty in terms of integrating semantic information through LLM fine-tuning, the limited performance improvements over simpler baselines raises questions about the practical value of this complexity. Second, the missing comparisons with recent relevant baselines like Meta TPP and LAMP is a significant limitation that makes it difficult to assess the true impact of the proposed approach. Finally, the lack of computational analysis and broader architectural studies suggests the work would benefit from more thorough investigation.

Given these considerations, particularly the inadequate baseline comparisons and limited performance improvements, I recommend rejection. While the authors made earnest attempts to address reviewer concerns, the current submission would benefit from substantial revision to address these fundamental limitations before meeting the conference's standards.

---

### Decision · Program_Chairs · 2025-01-22

Reject